# NeIF: Representing General Reflectance as Neural Intrinsics Fields for Uncalibrated Photometric Stereo

## Abstract

Uncalibrated photometric stereo (UPS) is challenging due to the inherent ambiguity brought by unknown light. Existing solutions alleviate the ambiguity by either explicitly associating reflectance to light conditions or resolving light conditions in a supervised manner. This paper establishes an implicit relation between light clues and light estimation and solves UPS in an unsupervised manner. The key idea is to represent the reflectance as four neural intrinsics fields, *i.e.*, position, light, specular, and shadow, based on which the neural light field is implicitly associated with light clues of specular reflectance and cast shadow. The unsupervised optimization of neural intrinsics fields can be free from training data bias and fully exploits all observed pixel values for UPS. Our method achieves a superior performance advantage over state-of-the-art UPS methods on public datasets and promising results under the challenging setting of sparse UPS. The code will be released soon.

## 1  Introduction

Photometric stereo (PS) [50] aims at recovering the surface normal from several light-varying images captured at a fixed viewpoint. As compared with other approaches (*e.g.*, multi-view stereo [41], active sensor-based solutions [58]), photometric stereo is excellent at recovering fine-detailed surfaces and has been widely used for Hollywood movies [11], industrial quality inspection [49], and biometrics [55]. Calibrating accurate lighting directions is crucial to the performance of photometric stereo methods [54]. However, lighting calibration is often tedious, dramatically restricting the applicability in the real-world. To this end, researchers develop uncalibrated photometric stereo (UPS) methods that estimate surface normal with unknown lights.

Uncalibrated photometric stereo suffers from General Bas-Relief (GBR) ambiguity [6] for an integrable surface. Early solutions address the ambiguity by explicitly associating reflectance to light, *i.e.*, adopting analytic reflectance models (*e.g.*, Lambertian reflectance [4], [37], parametric specular reflection [20], specular spikes [56], inter-reflection [12]) or imposing priors from reflectance properties [3, 25, 24, 44]. Thus, due to the strong reliance on reflectance assumption, these methods can be less effective for unknown reflectance. Besides, these methods ignore the clue of cast shadow and even fail in shadow regions due to the shadow-free surface assumption. Further, most of them require the light intensity to be identical for robust estimation. Recently, deep learning-based approaches address the ambiguity by estimating light direction and intensity before recovering surface normal [13, 15, 27, 40]. They train a light estimation network using a large-scale amount of rendered data in a supervised manner. However, the training data bias [33] can hardly be eliminated and can produce unexpected estimation for real-world data. Because rendered training data inevitably contains the domain gap from the real ones and scarcely cover all surfaces with different geometry and reflectance in the real-world. Besides, such a two-step solution can bring accumulating errors for surface normal

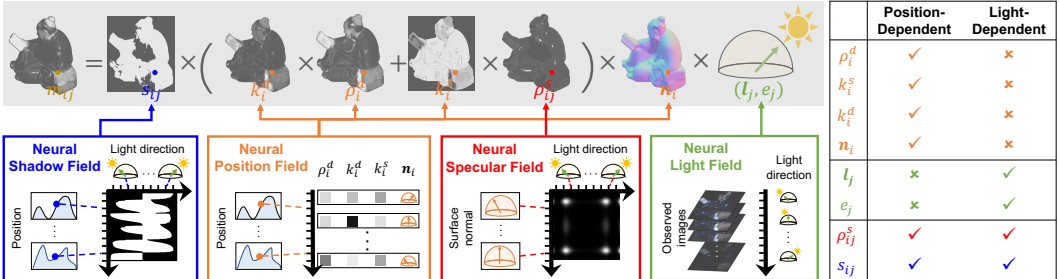

| | Position-Dependent | Light-Dependent |
|---|---|---|
| $\rho_i^d$ | ✓ | ✗ |
| $k_i^s$ | ✓ | ✗ |
| $k_i^d$ | ✓ | ✗ |
| $n_i$ | ✓ | ✗ |
| $l_j$ | ✗ | ✓ |
| $e_j$ | ✗ | ✓ |
| $\rho_{ij}^s$ | ✓ | ✓ |
| $s_{ij}$ | ✓ | ✓ |

Figure 1: Illustration of our neural intrinsics fields. Left-top: the rendering equation. Left-bottom: our four neural intrinsic fields, *i.e.*, from left to right: shadow, position, specular, and light fields, respectively. Each sub-figure in the left-bottom illustrates the mutual information across dimensions of position-light, position, normal-light, and observed images, respectively. Left figure shows how the neural intrinsics fields are imposed to render a pixel. Right: a summary of our intrinsics w.r.t. the dependence on position or light. The definition of notations can be found from Eq. (2) and Eq. (4).

estimation. Further, all these methods assume the light intensity distributed in a pre-defined range (*i.e.*, $[0.2, 2]$), restricting their applicability.

To this end, we propose **NeIF**, representing general reflectance as **N**eural **I**ntrinsics **F**ields for uncalibrated photometric stereo. Our method differs from previous methods in three aspects: 1) it fully considers clues of specular reflectance and cast shadow from each observed pixel for light estimation so that it is expected to produce accurate estimation for both light conditions and surface normal; 2) it does not make explicit assumptions about the reflectance or light so that it works with general surface reflectance and flexible light settings; 3) it infers light and surface normal in an unsupervised manner so that it is free from training data bias and achieves stable performance for data from different sources.

Our key idea is to represent the general reflectance as four neural intrinsics fields (*i.e.*, position, light, specular, and shadow, see Fig. 1), implemented by four multi-layer perceptrons (MLPs). These four fields are connected based on the implicit relation (or dependence) of these intrinsics so that no explicit assumption is imposed, *e.g.*, we take the estimated light as the input to recover specular reflectance and cast shadow instead of explicitly exploiting them for light estimation. These intrinsics fields are optimized to reconstruct each pixel value from observed images, which fully exploit mutual information across different dimensions, as shown in Fig. 1. The reconstruction error is backpropagated to the neural light field through neural specular and shadow fields so that clues of specular and shadow can be implicitly and fully considered for light estimation. Our contributions are summarized as:

- We represent general reflectance as four intrinsic neural fields to implicitly associate per-pixel reflectance to light, which solves uncalibrated photometric stereo by fully considering clues of specular reflectance and cast shadow for light estimation.

- We propose the NeIF, an uncalibrated photometric stereo method trained in an unsupervised manner, which works with general surface reflectance and flexible light settings, and is free from training data bias.

- We show that our method achieves superior performance over uncalibrated and unsupervised methods. We also demonstrate its excellent generalization capacity to data from different sources and promising performance with the challenging setup of sparse uncalibrated photometric stereo.

## 2 Related Work

This section mainly reviews the latest works in neural reflectance representation, and related works on unsupervised PS methods and UPS methods. Readers may refer to [45] and [13] for a more comprehensive summary.

### 2.1 Neural Reflectance Representation

Neural Radiance Fields (NeRFs) [34] focus on the 3D geometries without explicitly modeling the interaction between light and objects via the image formation model. Many subsequent works explore its application in various computer vision problems, such as relighting [16, 46], photometric stereo [29], and multi-view stereo [7, 9, 60, 28, 61]. These works require known lighting conditions [7, 46, 61, 29], adopt simple reflectance model [59], or leverage multi-view information [7, 9, 60, 28, 61]. Different from these methods, our method considers general reflectance, cast-shadow effects, and unknown light conditions for uncalibrated photometric stereo by taking images captured at a single viewpoint.

### 2.2 Unsupervised Photometric Stereo

Classical methods solve the calibrated photometric stereo problem without knowing the ground truth surface normal. Therefore, we classify them as unsupervised methods. The least square-based algorithm [50] provides the simplest solution, which assumes the object to be Lambertian. It is generally served as a baseline method due to its stability, but its strong assumption on the reflectance model makes it fail for non-Lambertian surface. The following works either regard the non-Lambertian reflectance components as the outliers [5, 18, 36, 51, 26, 52] or apply analytic reflectance models including Torrance-Sparrow [20], the Ward model [18], a mixture of multiple Ward models [21], [1] *etc.* to consider the non-Lambertian effects. However, the performance of those methods can only deal with limited types of materials. There are also more advanced methods that utilize the general reflectance features such as reciprocity, isotropy [2], and monotonicity [24]. Those methods give a reliable estimation for objects with a broad range of materials.

With the progress of deep learning, many learning-based frameworks have been proposed for calibrated photometric stereo. Taniai *et al.* [47] proposed the first unsupervised learning-based photometric stereo method through a rendering equation. However, their reflectance model does not separately consider shadow, specular highlights, and diffuse components.

We also train our method in an unsupervised manner. Different from previous methods, we address the challenging problem of UPS and separately model cast shadow, specular reflectance, and diffuse reflectance.

### 2.3 Uncalibrated Photometric Stereo

Previous works hold the Lambertian assumption and address the ambiguity brought by a $3\times3$ transformation matrix. Belhumeur *et al.* [6] reduce the dimension of the transformation to a three parameters GBR transformation by considering integrability constraints. Based on that, extra clues from reflectance, such as half-vector symmetry [31], albedo clustering [42], specular spikes [56], or assumptions of light source distribution, such as ring light [63], symmetry light [35], or uniform-distributed light sources [57, 4, 42, 53, 37, 31], are used to resolve the GBR ambiguity. However, all these methods require the light intensity to be identical, which is inapplicable in the real-world datasets such as DiLiGenT [45], Apple & Gourd [2], and Light Stage Data Gallery [11]. Cho *et al.* [17] put up a semi-calibrated method to deal with non-uniform light intensity, but they assume the light directions to be known. Quéau *et al.* [39] address the photometric stereo problem under inaccurate lighting calibration, while the accuracy can significantly drop when non-Lambertian components become dominant.

Recently, many deep learning methods have been proposed for uncalibrated photometric stereo. Chen *et al.* [13] propose a supervised uncalibrated framework, SDPS-Net, which can simultaneously estimate the light conditions (intensity and direction) and the surface normal. They suggest treating light estimation as a classification problem and separating the normal and light prediction to reduce the complexity. Their following work, GC-Net [15], improves the performance of SDPS-Net by

adding shading as an extra channel to the input of the light estimation network. However, as a common problem for all supervised methods, an over-fitting problem may occur due to the training data bias [33]. In contrast, unsupervised methods do not have such a concern. Another benefit is that there is no need to synthesize training sets for unsupervised network training. To utilize the advantage of unsupervised methods, Kaya *et al.* [27] propose a compromised method that trains the light estimation network in a supervised manner (similar to [13]), but estimate the surface normal in an unsupervised way (similar to [47]). However, they still suffer from training data bias during light estimation. Besides, all these methods make a strict assumption that the light intensity distributed in a pre-defined range (*i.e.*, [0.2, 2]) and suffers from accumulating error due to their two-step frameworks. In contrast, our method neither makes a strict assumption on reflectance nor needs special light source distribution and jointly solves light conditions and surface normal in an unsupervised manner.

## 3 Method

### 3.1 Problem Formulation

Given a set of observations $\boldsymbol{I} \triangleq (I_0, I_1, ..., I_f)$ of a static surface, illuminated by $f$ unknown directional illuminations distributing on the upper-hemisphere, uncalibrated photometric stereo aims at recovering light directions $\boldsymbol{L} \triangleq (\boldsymbol{l}_0, \boldsymbol{l}_1, ..., \boldsymbol{l}_f)$, light intensities $\boldsymbol{E} \triangleq (e_0, e_1, ..., e_f)$, and surface normal $\boldsymbol{N} \triangleq \{\boldsymbol{n}_i | i \in \mathbb{P}\}$, $\mathbb{P}$ is the set of all positions on the surface. The solution is achieved by solving the optimization problem,

$$\underset{\boldsymbol{L},\boldsymbol{E},\boldsymbol{N}}{\arg\min} \sum_{i=1}^{\#\mathbb{P}} \sum_{j=1}^{f} \mathrm{D}(\bar{m}_{ij}, m_{ij}), \tag{1}$$

where $\bar{m}_{ij} \in I_j$ is the observed pixel intensity at position $i$, $\#\mathbb{P}$ is the number of elements in $\mathbb{P}$[1], $m_{ij}$ is the corresponding rendered pixel intensity, $\mathrm{D}(\cdot, \cdot)$ is a metric describing their difference. Under an orthographic camera with linear radiometric response, $m_{ij}$ is formulated as (simplified in a per-pixel form),

$$m_{ij} = e_j \rho(\boldsymbol{n}_i, \boldsymbol{l}_j, \boldsymbol{v}) \max(\boldsymbol{n}_i^\top \boldsymbol{l}_j, 0) = e_j \rho_{ij} \max(\boldsymbol{n}_i^\top \boldsymbol{l}_j, 0), \tag{2}$$

where $\boldsymbol{v} = [0, 0, 1]$ is the view direction pointing toward the viewer, $\rho_{ij}$ describes the general reflectance, $\max(\boldsymbol{n}_i^\top \boldsymbol{l}_j, 0)$ represents the attach shadow.

Unknown light brings two ambiguities when solving Eq. (1), *i.e.*, shape-light ambiguity, which is denoted as an invertable matrix $\boldsymbol{G} \in \mathbb{R}^{3 \times 3}$, and reflectance-light ambiguity, which is denoted as a non-zero scalar $c_j \in \mathbb{R}$,

$$m_{ij} = e_j (c_j c_j^{-1}) \rho_{ij} \max(\boldsymbol{n}_i^\top (\boldsymbol{G} \boldsymbol{G}^{-1}) \boldsymbol{l}_j, 0). \tag{3}$$

### 3.2 Neural Intrinsics Fields

**General reflectance decomposition.** To exploit clues of specular reflectance and cast shadow for light estimation, we decompose the general reflectance as the cast shadow term $s_{ij}$ multiplying the bidirectional reflectance term,

$$\rho_{ij} = s_{ij}(k_i^d \rho_i^d + k_i^s \rho_{ij}^s), \tag{4}$$

where subscript '$i$' and '$j$' indicate position- (or normal-) and light-dependent factors, respectively; the cast shadow term $s_{ij}$ is either 0 or 1; $\rho_i^d, \rho_{ij}^s$ represent the diffuse and specular reflectance, and $k_i^d$ and $k_i^s$ are coefficients that balance out the effects of specular and diffuse reflectance[2].

**Neural fields of specular reflectance and cast shadow.** Previous methods identify specular features from *specific* pixels and associate an *explicit* relation to light for light estimation. However, this scheme fails for the surface where the specular features are invisible or the explicit relation is violated. Besides, leaving out the clue of cast shadow can obstruct producing competitive performance.

In contrast, we leverage *both* clues of specular and shadow for light estimation, which is achieved by building the neural specular field and neural shadow field, and associating the fields to light conditions.

---

[1]Without loss of generality, we put '#' before a set symbol to represent its number of elements in this paper.
[2]We think these coefficients of a point will not change under different lights, while they can be different at different positions.

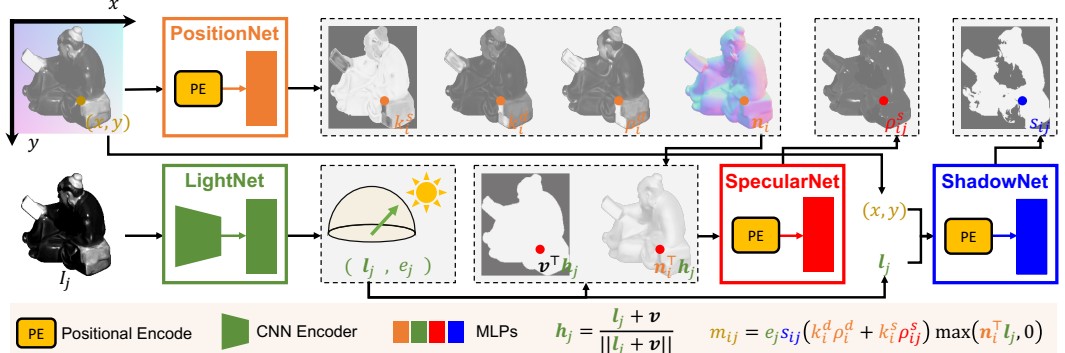

Figure 2: The framework of the proposed NeIF. PositionNet takes the input of positional code and outputs diffuse reflectance $\rho_i^d$, surface normal $\boldsymbol{n}_i$, and coefficients $k_i^d, k_i^s$. LightNet takes the observed image $I_j$ as the input and outputs light intensity $e_j$ and direction $\boldsymbol{l}_j$. SpecularNet takes $\boldsymbol{v}_j^\top \boldsymbol{h}_j$ and $\boldsymbol{n}^\top \boldsymbol{h}_j$ as the input and outputs specular reflectance $\rho_{ij}^s$. ShadowNet takes inputs of positional code and light direction $\boldsymbol{l}_j$ and output shadow indicator $s_{ij}$. All the intrinsics are used to render the observed pixel value $m_{ij}$ using Eq. (2) and Eq. (4).

These neural fields make the utmost of *all* observed pixels, and exploit the mutual information across different normal-light (specular) and position-light (shadow) for light estimation. Two MLPs, namely *SpecularNet* and *ShadowNet*, implement these neural fields, respectively. we take the estimated light direction as the input of these networks to achieve their *implicit* association to light conditions. Since the specular reflectance and cast shadow are normal- and position-dependent, we also feed the estimated surface normal and the positional code to them, respectively. SpecularNet and ShadowNet output $\rho_{ij}^s$ and $s_{ij}$, respectively, as shown in Fig. 2.

**Neural light field.** There is mutual information across different observed images, *i.e.*, observed images with similar appearances are expected to be illuminated by similar lights. To fully consider and exploit this mutual information, we build the neural light field by concatenating a CNN encoder to an MLP, namely *LightNet*. The encoder extracts a light code from each observed image[3], and the LightNet infers the corresponding light conditions (*i.e.*, $e_j, \boldsymbol{l}_j$) from the light code, as shown in Fig. 2.

**Neural position field.** There is mutual information across different positions on a surface, *i.e.*, the consistency of shape and diffuse reflectance in the spatial domain. To fully consider and exploit this mutual information, we establish a neural position field, namely *PositionNet*, implemented by an MLP. The neural position field outputs position-dependent, light-independent factors[4], *i.e.*, $\boldsymbol{n}_i, \rho_i^d, k_i^d, k_i^s$. The PositionNet takes the positional code as the input, as shown in Fig. 2.

### 3.3 Optimizing Neural Intrinsics Fields

We adopt the reconstruction loss function with the $\ell_1$ metric to optimize our NeIF,

$$\mathcal{L}_{\text{rec}} = \frac{1}{\#\mathbb{P} \times f} \sum_{i=1}^{\#\mathbb{P}} \sum_{j=1}^{f} |\bar{m}_{ij} - e_j s_{ij} (k_i^d \rho_i^d + k_i^s \rho_{ij}^s) \max(\boldsymbol{n}_i^\top \boldsymbol{l}_j, 0)|, \tag{5}$$

**Silhouette constraint.** $\mathcal{L}_{\text{rec}}$ cannot resolve the shape-light ambiguity in an unsupervised manner due to the inherently severe ill-posedness. Therefore, we introduce the silhouette constraint (similar to those in [23, 15]) to stabilize the training of PositionNet. To be specific, we use polynomial fitting with a moving window block to traverse and pre-compute the contour's normal of the given objects, represented as $\hat{\boldsymbol{N}}^{\text{si}} \triangleq \{\hat{\boldsymbol{n}}_k^{\text{si}} \in \mathbb{R}^{2 \times 1}, k \in \mathbb{S}\}$, $\mathbb{S}$ is the point set of the contour. We consider $\hat{\boldsymbol{N}}^{\text{si}}$ can

---

[3]We use light code instead of positional code because we experimentally find that the light code contains discriminative features of light intensity and direction.

[4]Since specular reflectance and cast shadow are both position-dependent and light-dependent, predicting them requires the input of light conditions, which increases the complexity of the neural position field. Therefore, we do not estimate them in the PositionNet, but predict them using SpecularNet and ShadowNet, respectively.

guide the prediction of the azimuth of boundary normal (at the same positions) and introduce the silhouette loss function,

$$\mathcal{L}_{\text{si}} = \sum_{k=1}^{\#\mathbb{S}} |\text{Nor}(\text{C}(\boldsymbol{n}_k)) - \hat{\boldsymbol{n}}_k^{\text{si}}|, \tag{6}$$

where $\boldsymbol{n}_k \in \mathbb{R}^{1\times3}$ represents the estimated surface normal at the positions of silhouette from PositionNet, $\text{C}(\cdot)$ cuts off the 3rd dimension of $\boldsymbol{n}_k$ (*i.e.*, $\text{C}(\boldsymbol{n}_k) \in \mathbb{R}^{1\times2}$), and $\text{Nor}(\cdot)$ is the vector normalization operation.

**Warm-up loss functions.** To avoid local minimum and achieve faster convergence, we warm up the NeIF in early-stage during training with three additional loss functions. We use the azimuth of lighting direction estimated by YS97 [57] to guide the training of LightNet,

$$\mathcal{L}_{\text{az}} = \frac{1}{f} \sum_{j=1}^{f} |\text{Nor}(\text{C}(\boldsymbol{l}_j)) - \text{Nor}(\text{C}(\boldsymbol{l}_j^{\text{az}}))|_2, \tag{7}$$

where $\boldsymbol{l}_j^{\text{az}}$ are estimated light directions by [57]. We adopt the gradient penalty [22] $\mathcal{L}_{\text{gp}}$ to stabilize the training of SpecularNet,

$$\mathcal{L}_{\text{gp}} = \frac{1}{\#\mathbb{P} \times f} \sum_{i=1}^{\#\mathbb{P}} \sum_{j=1}^{f} |\max(-\nabla_{\boldsymbol{n}_i^\top \boldsymbol{h}_j} \rho_{ij}^s, 0)|_2, \tag{8}$$

The intuition is from Blinn-Phong model [8], where the specular reflectance is monotonically increasing w.r.t the $\boldsymbol{n}_i^\top \boldsymbol{h}_j$, *i.e.*, $\nabla_{\boldsymbol{n}_i^\top \boldsymbol{h}_j} \rho_{ij}^s > 0$. We also supervise the training of ShadowNet using pseudo shadow maps $\hat{\boldsymbol{S}}_j, j = 1, 2, ..., f$,

$$\mathcal{L}_{\text{shadow}} = \frac{1}{\#\mathbb{P}} \sum_{j=1}^{} f|\boldsymbol{S}_j - \hat{\boldsymbol{S}}_j|_2, \tag{9}$$

The pseudo shadow maps are obtained by binarizing the observed images, *i.e.*, considering an observed pixel to be cast shadow if its intensity value is smaller than $0.2\times$ the mean intensity value of this image. After early-stage training, we discard these loss functions for a broad range of reflectance.

**NeIF training.** We train NeIF with the warm-up loss function in first 10 epochs,

$$\mathcal{L}_{\text{warmup}} = \mathcal{L}_{\text{rec}} + \lambda_{\text{si}}\mathcal{L}_{\text{si}} + \lambda_{\text{az}}\mathcal{L}_{\text{az}} + \lambda_{\text{gp}}\mathcal{L}_{\text{gp}} + \lambda_{\text{shadow}}\mathcal{L}_{\text{shadow}}, \tag{10}$$

where $\lambda_{\text{si}} = 5, \lambda_{\text{az}} = 0.1, \lambda_{\text{gp}} = 10, \lambda_{\text{shadow}} = 10$. We then train NeIF until 500 epochs or converging with the loss function,

$$\mathcal{L}_{\text{NeIF}} = \mathcal{L}_{\text{rec}} + \lambda_{\text{si}}\mathcal{L}_{\text{si}} + \lambda_{\text{shadow}}\mathcal{L}_{\text{recShadow}}. \tag{11}$$

where $\mathcal{L}_{\text{recShadow}}$ is the another shadow map supervision loss to train the ShadowNet. The loss function is the same to $\mathcal{L}_{\text{shadow}}$. The only difference is the calculation of $\hat{\boldsymbol{S}}_j$. For $\mathcal{L}_{\text{recShadow}}$, we calculate $\hat{\boldsymbol{S}}_j$ by rendering a depth map with the estimated $\boldsymbol{l}_j$. The depth map is reconstructed from the estimated surface normal map $\boldsymbol{N}$ by method [10]. $\mathcal{L}_{\text{recShadow}}$ is used to align outputs of ShadowNet to those of PositionNet.

**Implementation details.** We generate the positional code from the coordinate (in the image plane) by the same method in [34]. With the assumption of isotropic reflectance, we simplify the input of SpecularNet from $\{\boldsymbol{v}^\top \boldsymbol{l}_j, \boldsymbol{v}^\top \boldsymbol{n}_i, \boldsymbol{n}_i^\top \boldsymbol{l}_j\}$ to $\{\boldsymbol{v}^\top \boldsymbol{h}_j, \boldsymbol{n}_i^\top \boldsymbol{h}_j\}$[5] [31] for easier training. Similar to the Cook-Torrance reflectance model [19], we assume $k_i^d + k_i^s = 1$ to reduce the number of unknowns. The CNN decoder with declining channels processes the down-sampled images with a dimension of $256 \times 256$. The LightNet takes flatten features to predict $\boldsymbol{l}_j, e_j$ in two different branches. For the ShadowNet, we also generate the positional code for $\boldsymbol{l}_j$ and concatenate it to the feature in the 9th layer. The output of ShadowNet is either 0 or 1, which is realized by a step function with a similar implementation in [38].

---

[5]$\boldsymbol{h}_j$ is the bisector of $\boldsymbol{l}_j$ and $\boldsymbol{v}$, $\boldsymbol{h}_j = \frac{\boldsymbol{l}_j + \boldsymbol{v}}{\|\boldsymbol{l}_j + \boldsymbol{v}\|}$.

Table 1: Quantitative comparison in terms of mean angular error for surface normal on DiLiGenT benchmark [45]. This table summarizes comparison methods. 'N.A.' represents not applicable as calibrated PS is with known $\ell$ and $e$. 'Semi' indicates certain method leverage partial information of light. '✓' (or '✗') represents that certain methods (do not) adopt supervised learning for the estimation of surface normal $n$, light direction $l$, or light intensity $e$. 'Identical' means certain methods require the light intensity of different illuminating to be identical.

| Method | PS/ UPS | $n$ Supervision | $l$ Supervision | $e$ Supervision | BALL | BEAR | BUDDHA | CAT | COW | GOBLET | HARVEST | POT1 | POT2 | READING | AVG |
|---|---|---|---|---|---|---|---|---|---|---|---|---|---|---|---|
| LS [50] | PS | ✗ | N.A. | N.A. | 4.10 | 8.39 | 14.92 | 8.41 | 25.60 | 18.50 | 30.62 | 8.89 | 14.65 | 19.80 | 15.39 |
| TM18 [47] | PS | ✗ | N.A. | N.A. | 1.47 | 5.79 | 10.36 | 5.44 | 6.32 | 11.47 | 22.59 | **6.09** | 7.76 | **11.03** | 8.83 |
| YS97 [57] | UPS | ✗ | ✗ | Identical | 39.12 | 41.30 | 43.02 | 39.10 | 47.18 | 42.25 | 79.56 | 42.94 | 41.88 | 41.06 | 45.74 |
| AM07 [4] | UPS | ✗ | ✗ | Identical | 7.27 | 16.81 | 32.81 | 31.45 | 54.72 | 46.54 | 61.70 | 18.37 | 49.16 | 53.65 | 37.25 |
| SM10 [42] | UPS | ✗ | ✗ | Identical | 8.90 | 11.98 | 15.54 | 19.84 | 22.73 | 48.79 | 73.86 | 16.68 | 50.68 | 26.93 | 29.59 |
| WT13 [53] | UPS | ✗ | ✗ | Identical | 4.39 | 6.42 | 13.19 | 36.55 | 19.75 | 20.57 | 55.51 | 9.39 | 14.52 | 58.96 | 23.93 |
| LM13 [32] | UPS | ✗ | ✗ | Identical | 22.43 | 15.44 | 25.76 | 25.01 | 22.53 | 29.16 | 34.45 | 32.82 | 20.57 | 48.16 | 27.63 |
| PF14[37] | UPS | ✗ | ✗ | Identical | 4.77 | 9.07 | 14.92 | 9.54 | 19.53 | 29.93 | 29.21 | 9.51 | 15.90 | 24.18 | 16.66 |
| LC17 [31] | UPS | ✗ | ✗ | Identical | 9.30 | 10.90 | 19.00 | 12.60 | 15.00 | 18.30 | 28.00 | 12.40 | 15.70 | 22.30 | 16.35 |
| CH18 [14] | Semi | ✓ | ✗ | Known | 3.96 | 7.19 | 13.06 | 12.16 | 11.84 | 18.07 | 27.22 | 11.13 | 11.11 | 20.46 | 13.62 |
| CH19 [13] | UPS | ✓ | ✓ | ✓ | 2.77 | 6.89 | 8.97 | 8.06 | 8.48 | 11.91 | 17.43 | 8.14 | 7.50 | 14.90 | 9.51 |
| CW20 [15] | UPS | ✓ | ✓ | ✓ | 2.50 | 5.60 | **8.60** | 7.90 | 7.80 | 9.60 | **16.20** | 7.20 | 7.10 | 14.90 | 8.71 |
| CM20 [17] | Semi | ✗ | Known | ✗ | 2.78 | 8.07 | 13.38 | 8.05 | 26.90 | 18.18 | 33.35 | 9.47 | 19.58 | 14.19 | 15.40 |
| KK21 [27] | UPS | ✓ | ✓ | ✓ | 3.78 | 5.96 | 13.14 | 7.91 | 10.85 | 11.94 | 25.49 | 8.75 | 10.17 | 18.22 | 11.62 |
| SK22 [40] | UPS | ✓ | ✓ | ✓ | 3.46 | 5.48 | 10.00 | 8.94 | **6.04** | 9.78 | 17.97 | 7.76 | 7.10 | 15.02 | 9.15 |
| **Ours** | UPS | ✗ | ✗ | ✗ | **1.15** | **4.41** | 8.78 | **5.08** | 6.14 | **9.49** | 17.68 | 7.94 | **6.12** | 11.82 | **7.86** |

## 4 Experiments

**Training details.** Our main framework is implemented in PyTorch, while the pre-calculations (method [57] and silhouette fitting) are implemented in MATLAB. We use Adam as the optimizer with a learning rate $\alpha = 5 \times 10^{-4}$ to train our framework in 500 epochs for each scene separately, and the warm-up stage takes up 10 epochs. The last 100 epochs use a lower learning rate $\alpha = 5 \times 10^{-5}$ for fine-tunning. The batch size is 32 for lighting and 256 for spatially random sampling. After every epoch, the depth map is reconstructed according to the predicted normal map by method [10]. Each scene takes from 2 hours to 6 hours on one RTX 2080Ti 12GB GPU, depending on the resolution of the objects.

**Evaluation metrics.** We adopt the same metric in [13], the scale-invariant relative error, to measure the accuracy of recovered light intensity as,

$$E_{\text{int}} = \frac{1}{f} \sum_{j=1}^{f} \left( \frac{|\eta e_j - \tilde{e}_j|}{\tilde{e}_j} \right). \tag{12}$$

where, $\eta$ is calculated by solving $\arg\min_\eta \sum_{j=1}^{f} (\eta e_j - \tilde{e}_j)^2$ by least squares minimization. The metric to measure the accuracy of the predicted light directions and surface normal is the widely used mean angle error (MAE) in degree.

### 4.1 Evaluation on Public Datasets

Since the proposed NeIF is an unsupervised uncalibrated photometric stereo method, we compare its performance with state-of-the-art uncalibrated and unsupervised photometric stereo methods. Three real-world datasets, including the DiLiGenT benchmark dataset [45], APPLE & GOURD dataset [2], and LIGHT STAGE DATA GALLERY dataset [11], are used for evaluation.

**Quantitative comparison for normal map.** Table 1 lists relevant works for a comprehensive surface normal estimation comparison. As summarized in Table 1, our method is the only method that addresses UPS without the supervision of $N$, $L$, or $E$. However, our method achieves the best performance and maintains a considerable advantage over other competitors [13, 15] (*e.g.*, 7.86 for NeIF vs. 8.71 for [15]). Handling the objects in DiLiGenT dataset [45] with different shapes and reflectance, numbers from our method are either best or competitive (about $1°$ as compared with the best performing UPS method), which shows its good generalization capacity to general reflectance and various shapes. This is because our NeIF fully considers mutual information by building up intrinsics fields and the implicit modeling facilitates general reflectance modeling.

**Visual quality comparison for the normal map.** Fig. 3 illustrates the visual quality comparison in terms of recovered surface normal maps and corresponding error maps on DiLiGenT dataset [45].

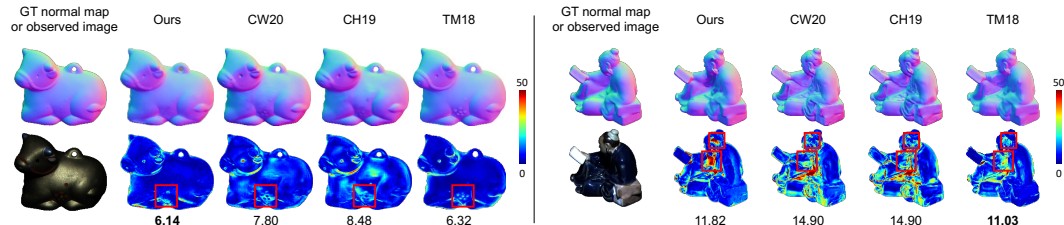

Figure 3: Visual quality comparison in terms of normal map and error map on COW (left) and READING (right) from DILIGENT [45]. For each subfig, from left to right: ground truth of normal map or observed image, normal map / error map from our method, CW20 [15], CH19 [13], and TM18 [47].

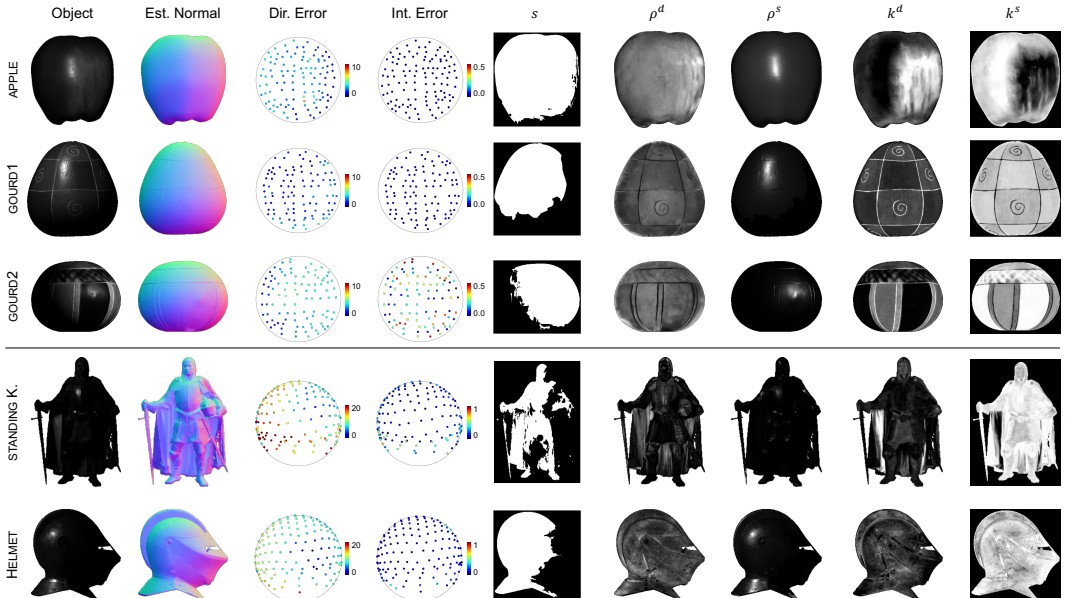

Figure 4: From left to right: the reference image, estimated surface normal map from our method, the lighting direction error, light intensity error, the estimated shadow map, diffuse reflectance map, specular reflectance map, diffuse scaling coefficient map, and specular coefficient map. Top three objects are APPLE, GOURD1, and GOURD2 from APPLE & GOURD dataset [2], and bottom two objects are STANDING KNIGHT and HELMET from LIGHT STAGE DATA GALLERY dataset [11].

The comparison is conducted with two state-of-the-art UPS methods [13, 15] and a state-of-the-art unsupervised PS method [47]. Our method is less sensitive to spatially-varying albedo due to the per-pixel manner (red boxes in the left subfigure of Fig. 3). However, although the positional code considers the global shape effect, this per-pixel manner is less effective in modeling complex shape information as compared with the all-pixel one. It fails for regions with cast shadow (or overexposure) under most light directions (red boxes in the right subfigure of Fig. 3). We also show the visual quality results for APPLE & GOURD dataset [2] and the LIGHT STAGE DATA GALLERY dataset [11] in Fig. 4. Our method produces reliable estimation for most regions, thanks to the full exploitation of mutual information across different dimensions.

**Quantitative comparison for light directions and intensities.** As can be observed in Table 2 and Table 3, our method achieves a superior performance advantage over unsupervised methods (YS97 [57] and PF14 [37]) while maintaining competitive performance as compared with supervised methods (CH19 [13] and CW20 [15]). These supervised methods adopt two-step solutions and suffer from accumulating error. Therefore, even though they achieve similar performance in terms of light conditions accuracy, they are less effective on estimating surface normal as compared with our method (see Table 1).

Table 2: Quantitative comparison in terms of mean angluar error for light direction and scale-invariant error for intensity on DiLiGenT benchmark [45].

| Model | BALL dir. | int. | BEAR dir. | int. | BUDDHA dir. | int. | CAT dir. | int. | COW dir. | int. | GOBLET dir. | int. | HARVEST dir. | int. | POT1 dir. | int. | POT2 dir. | int. | READING dir. | int. | AVG dir. | int. |
|---|---|---|---|---|---|---|---|---|---|---|---|---|---|---|---|---|---|---|---|---|---|---|
| YS97 [57] | 12.41 | 0.334 | 14.06 | 0.260 | 11.68 | 0.300 | 13.75 | 0.318 | 15.79 | 0.251 | 15.24 | 0.316 | 59.41 | 0.586 | 12.99 | 0.322 | 12.58 | 0.283 | 13.08 | 0.266 | 18.10 | 0.320 |
| PF14 [37] | 4.90 | 0.036 | 5.24 | 0.098 | 9.76 | 0.053 | 5.31 | 0.059 | 16.34 | 0.074 | 33.22 | 0.223 | 24.99 | 0.156 | 2.43 | **0.017** | 13.52 | 0.044 | 21.77 | 0.122 | 13.75 | 0.088 |
| CH19 [13] | 3.27 | 0.039 | 3.47 | 0.061 | 4.34 | 0.048 | 4.08 | 0.095 | 4.52 | 0.073 | 10.36 | 0.067 | 6.32 | 0.082 | 5.44 | 0.058 | 2.87 | 0.048 | **4.50** | 0.105 | 4.92 | 0.068 |
| CW20 [15] | 1.75 | **0.027** | **2.44** | 0.101 | 2.86 | **0.032** | 4.58 | 0.075 | **3.15** | **0.031** | 2.98 | 0.042 | **5.74** | 0.065 | **1.41** | 0.039 | 2.81 | 0.059 | 5.47 | 0.048 | 3.32 | 0.052 |
| **Ours** | **1.69** | 0.030 | 3.96 | **0.010** | **1.73** | 0.032 | **2.92** | **0.021** | 4.98 | 0.050 | 6.82 | **0.040** | 7.06 | **0.032** | 3.33 | 0.134 | **3.71** | **0.028** | 7.45 | **0.042** | 4.37 | **0.042** |

Table 3: Quantitative comparison in terms of mean angular error for light direction and scale-invariant error for intensity on APPLE & GOURD [2] and LIGHT STAGE DATA GALLERY [11].

| Model | APPLE dir. | int. | GOURD1 dir. | int. | GOURD2 dir. | int. | AVG dir. | int. | STANDING KNIGHT dir. | int. | HELMET dir. | int. | AVG dir. | int. |
|---|---|---|---|---|---|---|---|---|---|---|---|---|---|---|
| YS97 [57] | 25.71 | 0.400 | 22.23 | 0.329 | 29.30 | 0.347 | 25.75 | 0.359 | 37.48 | 0.533 | 34.43 | 0.476 | 35.96 | 0.505 |
| PF14 [37] | 6.68 | 0.109 | 21.23 | 0.096 | 25.87 | 0.329 | 17.92 | 0.178 | 33.81 | 1.311 | 25.40 | 0.576 | 29.61 | 0.944 |
| CH19 [13] | 9.31 | 0.106 | 4.07 | 0.048 | 7.11 | **0.186** | 6.83 | 0.113 | 11.60 | 0.286 | 6.57 | 0.212 | 9.09 | 0.249 |
| CW20 [15] | 10.91 | 0.094 | 4.29 | 0.042 | 7.13 | 0.199 | 7.44 | 0.112 | **5.31** | 0.198 | **5.33** | 0.096 | **5.32** | 0.147 |
| **Ours** | **2.65** | **0.011** | **1.76** | **0.029** | **3.21** | 0.230 | **2.54** | **0.090** | 13.38 | **0.189** | 8.12 | **0.082** | 10.75 | **0.135** |

## 4.2 Evaluation on Sparse Uncalibrated Photometric Stereo

As compared with other per-pixel PS methods (*e.g.*, ZJ19 [62] and LL19 [30]), the proposed NeIF exploits much more constraints to estimate the normal at a point (*i.e.*, $\#\mathbb{P} \times f$ vs. $f$). Because it learns fields instead of regressing intensity profile [43] to surface normal. Therefore, we investigate the effectiveness of our method under the challenging setting of sparse UPS.

We randomly select 10 or 16 images illuminated by different lights from DiLiGenT dataset [45] and test our method using these images. We repeat this process 30 times, similar to [62]. We compare the performance with that from the classical UPS method PF14 [37], a deep learning based per-pixel approach ZJ19 [62], and an all-pixel approach CH18 [14]. Due to the fewer reconstruction terms to train our NeIF, we slightly increase the dimension of the positional encoding module from 4 to 6 that increases the frequency [48] to stabilize the training, *i.e.*, strengthening the role of positional code to reduce the variance of estimated intrinsics. Besides, we drop $\mathcal{L}_{az}$ during early stage warm-up because YS97 [57] fails for the sparse inputs.

We report the mean results over 30 trails in Table 4. As can be observed, our method achieves competitive performance on normal estimation as compared with state-of-the-art sparse PS methods with known light (*i.e.*, 11.62 for NeIF vs. 9.82 for ZJ19 [62] under 10 lights, and 10.38 for NeIF vs. 9.00 for CH18 [14] under 16 lights). Kindly note that we do not require any ground truth normal for supervision. As compared with the unsupervised UPS method (PF14 [37]), we achieve a superior performance advantage. The results validate the effectiveness of building up neural intrinsics fields that fully exploit the mutual information across different dimensions.

## 5 Conclusion

This paper proposes NeIF for UPS. By representing the general reflectance as four neural intrinsics fields, it implicitly imposes the light clues of specular reflectance and cast shadow for light estimation, which facilitates solving UPS with general reflectance. The proposed NeIF can fully exploit mutual information from all observed pixel values so that it produces stable estimation for PS and sparse PS. The unsupervised training manner of NeIF is beneficial to the generalization capacity of data from different sources.

**Limitations.** Although our method produces promising results for light conditions and surface normal estimation, it has several limitations. First, the estimated intrinsics, such as shadow, specular and diffuse reflectance, and their balancing coefficients are less accurate for objects with complicated geometries, as shown in Fig. 4. Second, we need to balance the role of positional code and estimated intrinsics when applying our method for sparse UPS. However, how to find an optimal balancing strategy is unknown. Third, as shown in Fig. 3, our method is less effective on global effects, especially for regions with cast shadow (or overexposure) under most light directions due to our

Table 4: Quantitative comparison in terms of mean angular error for surface normal on DɪLɪGєɴT dataset [45]. We compare our method with supervised PS methods (CH18 [14] and ZJ19 [62]), and unsupervised UPS methods (PF14 [37]) under different randomly selected lights. All of the selected methods are retested under the same light setting with us. In the table, '$(x)$' indicates $x$ lights are used in the certain methods; '✓' (or '✗') represents that certain methods (do not) adopt supervised learning for the estimation of surface normal $n$ (note that light information is known in CH18 [14] and ZJ19 [62]). Bold font indicates the best performance under 10 lights and 16 lights, respectively.

| Model | PS/UPS | Supervision | BALL | BEAR | BUDDHA | CAT | COW | GOBLET | HARVEST | POT1 | POT2 | READING | AVG |
|---|---|---|---|---|---|---|---|---|---|---|---|---|---|
| CH18 (10) [14] | PS | ✓ | 3.87 | 6.35 | **9.20** | 8.47 | 9.95 | 10.68 | 18.40 | 9.01 | 8.97 | 15.29 | 10.02 |
| ZJ19 (10) [62] | PS | ✓ | 4.38 | 5.79 | 9.60 | **7.13** | **7.87** | **10.00** | 18.35 | **8.41** | 11.20 | **15.45** | **9.82** |
| PF14 (10) [37] | UPS | ✗ | 69.95 | 10.19 | 17.47 | 11.11 | 22.23 | 49.77 | 46.96 | 10.38 | 19.69 | 31.50 | 28.93 |
| **Ours (10)** | UPS | ✗ | **2.49** | **4.90** | 11.58 | 7.47 | 10.21 | 14.45 | 27.30 | 11.41 | **6.57** | 19.85 | 11.62 |
| CH18 (16) [14] | PS | ✓ | 3.27 | 5.98 | **8.48** | **7.21** | 8.58 | **9.48** | **17.04** | 8.06 | 8.15 | **13.73** | **9.00** |
| PF14 (16) [37] | UPS | ✗ | 63.76 | 9.15 | 15.24 | 9.13 | 20.30 | 41.52 | 33.58 | 9.78 | 17.08 | 26.40 | 24.60 |
| **Ours (16)** | UPS | ✗ | **2.34** | **4.52** | 10.33 | 7.87 | **7.68** | 11.23 | 25.07 | **8.00** | **6.35** | 20.42 | 10.38 |

per-pixel manner. Besides, the inference of our method is inefficient due to the unsupervised training manner and the per-pixel estimation, which is inapplicable for real-time applications.

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
