# Supplementary Material for Rebuttal of "NeIF: Representing General Reflectance as Neural Intrinsics Fields for Uncalibrated Photometric Stereo"

In this supplementary material,

1. Sec. 12: we collect data in casual scenarios to validate the effectiveness of our method on unsupervised learning and light-normal joint optimization.

2. Sec. 13: we visualize the BRDFs of different points to demonstrate that our method could generate spatially varying BRDF.

3. Sec. 14: we list the testing time of our method for objects in DiLiGenT benchmark [8].

4. Sec. 15: we compare our method with those that remove the ShadowNet and use the shadow maps obtained from simple binarization of observed images (as in [6]).

5. Sec. 16: we provide an intuitive explanation of how silhouette constraints work in our method.

6. Sec. 17: we randomly scale the intensity of input images and provide an experimental comparison.

7. Sec. 18: we revise the numbers of Table 2 in the main paper (indicated by red color). We also provide the standard deviation and the mean value for 10 objects on the DiLiGenT benchmark [8] under 5 random tests.

8. Sec. 19: we switch our rendering model to be identical to [6], where we use 9 specular basis instead of 1.

## 12 Comparison on Data Collected under Casual Environments

To further highlight the advantage of our method regarding unsupervised learning and light-normal joint optimization, we perform an experimental comparison on data collected in casual environments.

**Captured objects.** We use three objects for this experiment, including Bunny, Venus, and Mouse. Bunny contains lots of fine details, with a broad specular lobe on a uniform material (phenolic resin). Venus is made up of glass for the pearl on the tray and gypsum for the body. Mouse has lots of defects on the shell, made up of a spatially varying but mostly diffuse material. The illustration of these objects can be found in Fig. 21.

**Data capture**. We use the iPhone 13 Pro Max camera fixed on a phone tripod for data capture. We turn the HDR mode on and set the exposure compensation to -2 to avoid overexposure. We record a video with a raw resolution at $1920 \times 1080$. During video recording, we move the light source slowly to illuminate the object at different angles and try the keep the trajectory parallel to the image plane. The captured video could be found at anonymous link[1]. Fig. 20 shows the capture equipment.

---

[1] https://drive.google.com/drive/folders/1XZEb2H3-ZTuTyZaxiewa3DDH-urVbuOk?usp=sharing

Submitted to 36th Conference on Neural Information Processing Systems (NeurIPS 2022). Do not distribute.

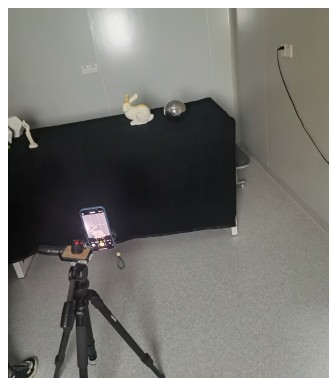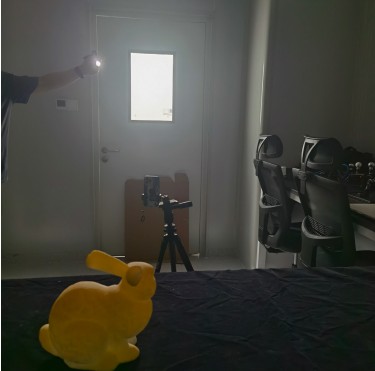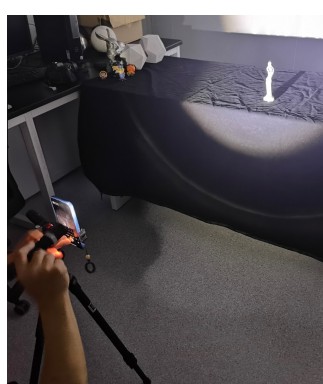

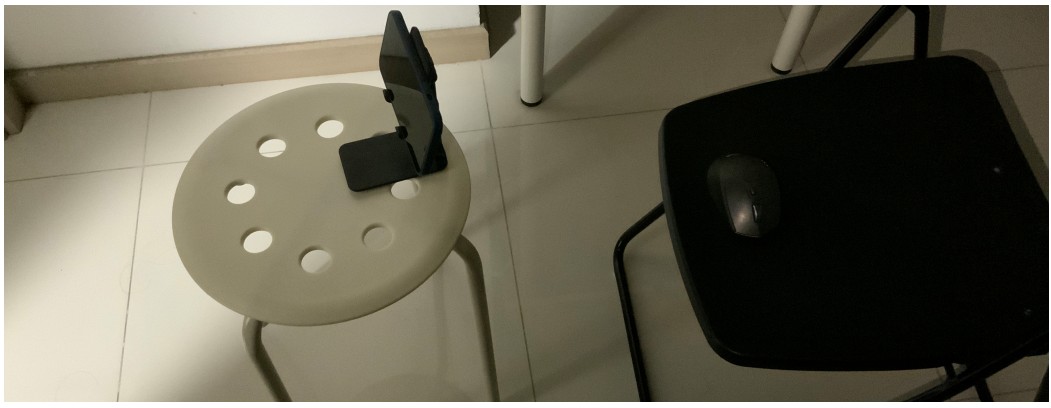

Figure 20: Data collection process. From top to the bottom, Row 1: the equipment we use in scene 1 to capture the VENUS and BUNNY; Row 2: the equipment we capture MOUSE in scene 2.

BUNNY and VENUS are shot at about 1 meter away with the light at about 1-1.5 meters away. MOUSE is shot at about 30cm away with light at about 40cm away.

**Ambient light and light sources**. All data are captured with ambient light because the full-dark chamber is also expensive for casual users. For the data capture of BUNNY and VENUS, we control the impact of the ambient light by changing the intensity of the electric torch, i.e., weak impact (or strong light source intensity) for BUNNY, and strong impact (or weak light source intensity) for VENUS. The capture environment for these objects is relatively controlled, and the ambient light is from a small window (see Fig. 20). For the data capture of MOUSE, we consider a more casual and challenging scenario (see Fig. 20). That is, the ambient light is more dominant and uncontrolled. We use a flashlight from the mobile phone to further relax the assumption of directional light in photometric stereo.

**Data processing.** We use MATLAB 2020 to extract 100 frames uniformly from each video and further downsample them to the resolution of $960 \times 540$. We extract the mask of each object via PhotoShop 2020.

**Comparison methods.** We compare our method with CW20 [4]. As LL22 [6] is not for UPS, we feed it by the estimated light from the state-of-the-art method CW20 [4] and denoted it as CW20 [4]+LL22 [6].

### 12.1 Comparison of Surface Normal

The estimated normal map for each object is shown in Fig. 21. However, CW20 [4] and CW20 [4]+LL22 [6] is sensitive to the data bias in supervised learning of light estimation model. The error of estimated light dramatically degrades the performance of normal estimation. In contrast, our method is robust to casual environments. We have a more reasonable estimation on BUNNY with the ambient light and MOUSE in the challenging scene and much better results on VENUS as

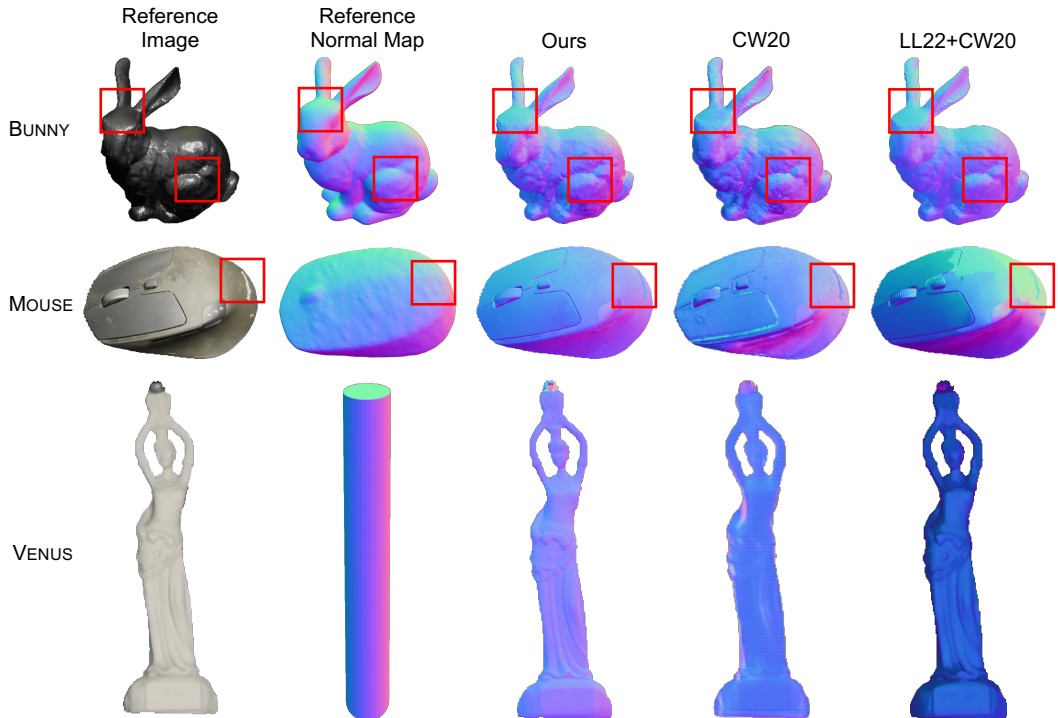

Figure 21: Visual quality comparison in terms of normal map on BUNNY (row 1), MOUSE (row 2), and VENUS (row 3) from DILIGENT [8]. For each subfig, from left to right: mean of the observed images, coarse normal map download in `https://sketchfab.com/`, that has similar shape with the objects for reference, normal map from our method, CW20 [4], LL22 [6] + CW20 [4].

compared with CW20 [4] and CW20 [4]+LL22 [6]. Since our model adopts the directional light assumption, it produces inaccurate light. The superior performance advantage on surface normal estimation indicates that our method could better balance the accuracy of the surface normal and light due to its light-normal joint optimization manner.

## 12.2 Comparison of Light Estimation

We further make a comparison on the estimation of light. The predicted light direction projected on the XY-plane is shown in Fig. 22 and represents the estimated light intensity by color (note that the light intensity is normalized for better visualization). Since the intensity of our light source during data collection is unchanged, our method produces much more accurate light intensity than the comparison methods. Besides, we change the light direction by moving the light source at a large distance. The trajectory recovered by our method is more reasonable than the comparison methods, especially for objects of MOUSE and VENUS. These results indicate that our method is more robust to training data bias due to its unsupervised manner.

## 13 Spatially Varying BRDF

We visualize the BRDF in different position of KNIGHT STANDING in Fig. 23. Although we are restricted by our specular model, we can still generate spatially varying BRDF by assigning different scaling factors $k_d \in [0, 1]$ to different spatial points. That is, $k_d$ controls the specularity of a certain point. The reflectance could be Lambertion if $k_d$ is 1 and non-Lambertian otherwise. For instance, we have a larger $k_d$ on the knight's face and cloak, but a smaller $k_d$ on the armor, generating different BRDF and shading effects shown in Fig. 23.

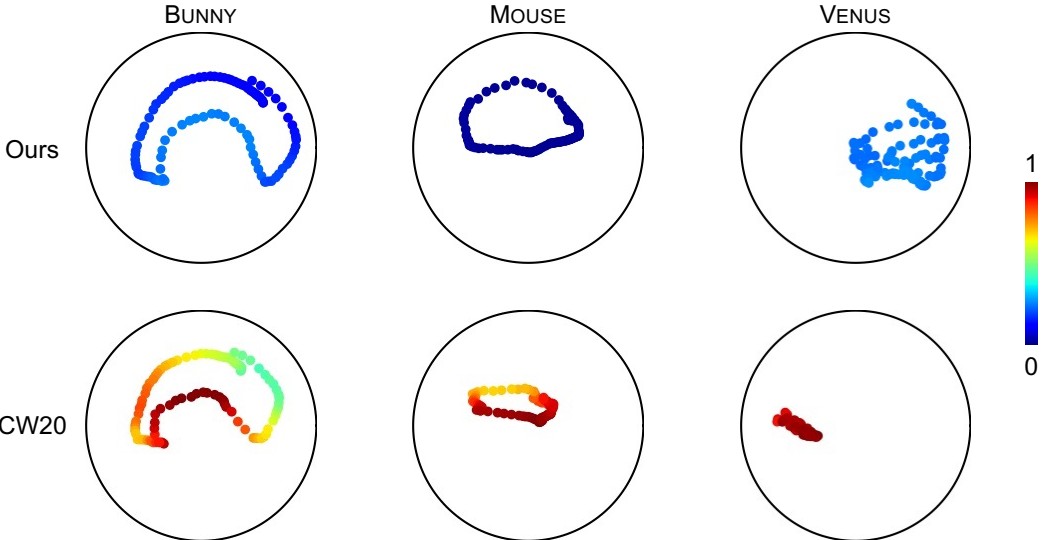

Figure 22: Visual comparison in terms of the estimated light trajectory on BUNNY(column 1), MOUSE (column 2), and VENUS (column 3). For each sub-figure, from top to bottom: Ours, CW20 [4]'s predicted light trajectory that projected onto the XY-plane, the color indicates the value of the light intensity.

## 14 Test Time on DILIGENT Benchmark [8]

KK21 [5] did not release their code. Therefore, we report the test time (s) for TM18 [9], LL22 [6] and ours in Table 10 under different numbers of input, respectively. Our method has the fastest test time given 96 images as input and competitive test time given 16 images as input among the mentioned methods. The tests are implemented on an RTX3090 GPU, and the batch size is full size for lighting and 2048 for spatially sampling. The larger the batch size for spatially sampling, the faster the test, but the GPU memory occupation will also increase. According to our experiments, the parameters we choose are the most cost-efficient, which will occupy around 3.2GB on RTX3090.

Table 10: Quantitative comparison in terms of test time (s) on DILIGENT benchmark [8]. Numbers in brackets indicate the number of lights. For example, Ours (96) indicates that our method takes 96 images under different lights as the input.

| | BALL | BEAR | BUDDHA | CAT | COW | GOBLET | HARVEST | POT1 | POT2 | READING | AVG |
|---|---|---|---|---|---|---|---|---|---|---|---|
| TM18 (96) [9] | **0.21** | **0.70** | **0.79** | 1.02 | **0.48** | 1.07 | 1.07 | 1.18 | 0.84 | 0.54 | 0.79 |
| LL22 (96) [6] | 1.22 | 0.92 | 1.17 | 1.36 | 0.85 | 1.26 | 1.23 | **1.08** | 1.14 | 1.06 | 1.13 |
| Ours (96) | 0.29 | **0.70** | 0.83 | **0.82** | 0.50 | **0.46** | **1.04** | 1.10 | **0.65** | **0.52** | **0.69** |
| TM18 (16) [9] | **0.04** | **0.11** | **0.12** | **0.17** | **0.08** | **0.22** | **0.17** | **0.17** | **0.14** | **0.13** | **0.13** |
| LL22 (16) [6] | 0.62 | 0.69 | 0.69 | 0.68 | 0.67 | 0.66 | 0.71 | 0.66 | 0.69 | 0.77 | 0.68 |
| ous (16) | 0.27 | 0.67 | 0.80 | 0.83 | 0.49 | 0.49 | 1.08 | 1.09 | 0.63 | 0.52 | 0.69 |

## 15 Statistical Shadow Handling

We remove the ShadowNet and train our method using the pseudo shadow maps for reconstruction loss. The pseudo shadow maps are obtained by binarizing the observed images, *i.e*, considering an observed pixel to be cast shadow if its intensity value is smaller than $0.2\times$ the mean intensity values of this image. The result is shown in Table 11. Although pseudo shadow map works well on objects like BALL, BUDDHA, and CAT, the performance, the accuracy of the light directions, dropped significantly on objects like HARVEST, GOBLET, and READING. This is because pseudo shadow maps do not provide any extra clues for the LightNet through the back-propagation.

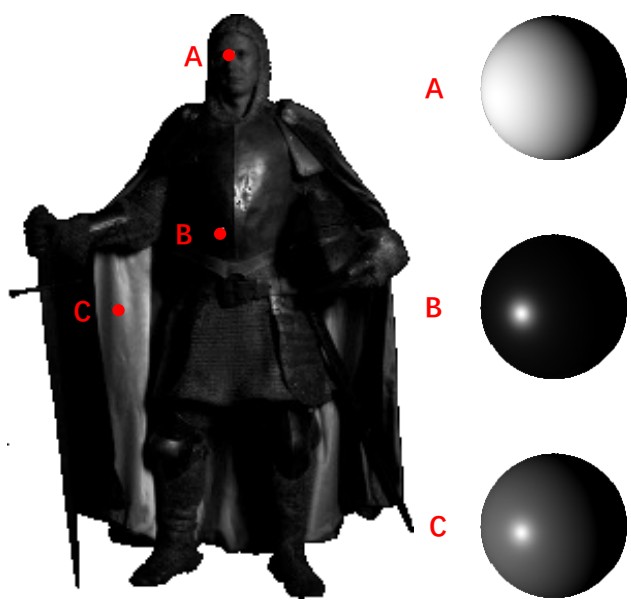

Figure 23: Visual illustration of BRDF at different positions. On the left is the observed image of STANDING KNIGHT. We select three different points on the object and show our predicted BRDF spheres of those points on the right side. We normalized the image and the predicted BRDF from 0 to 1 for a better illustration. For the right, sub-figures from top to bottom are: BRDF for the point on the face, BRDF for the point on the armor, BRDF for the point on the cloak, respectively.

Table 11: Quantitative comparison in terms of normal map error, light direction error, and light intensity error on DILIGENT dataset [8]. 'Ours w $\hat{s}$' indicates using pseudo shadow map instead of the ShadowNet for shadow handling.

|  |  | BALL | BEAR | BUDDHA | CAT | COW | GOBLET | HARVEST | POT1 | POT2 | READING | AVG |
|---|---|---|---|---|---|---|---|---|---|---|---|---|
| Ours w $\hat{s}$ | norm. | 1.68 | 6.52 | 8.33 | 4.87 | 7.92 | 18.32 | 39.08 | 8.02 | 8.20 | 12.91 | 11.59 |
|  | dirs | 2.60 | 5.00 | 2.32 | 1.34 | 6.82 | 26.61 | 14.48 | 9.28 | 6.69 | 12.25 | 8.74 |
|  | ints | 0.014 | 0.012 | 0.030 | 0.024 | 0.067 | 0.056 | 0.530 | 0.037 | 0.026 | 0.060 | 0.086 |
| Ours | norm. | 1.15 | 4.41 | 8.78 | 5.08 | 6.14 | 9.49 | 17.68 | 7.94 | 6.12 | 11.82 | 7.86 |
|  | dir. | 1.69 | 3.96 | 1.73 | 2.92 | 4.98 | 6.82 | 7.06 | 3.33 | 3.71 | 7.45 | 4.37 |
|  | int. | 0.030 | 0.010 | 0.032 | 0.021 | 0.050 | 0.040 | 0.032 | 0.134 | 0.028 | 0.042 | 0.042 |

## 16  Silhouette Constraint

For Lambertian objects, the GBR ambiguity is represented as:

$$\boldsymbol{I}^k = \boldsymbol{B}^\top \boldsymbol{S}^k. \tag{1}$$

Where, $\boldsymbol{G}$ is the $3 \times 3$ ambiguity matrix; $\boldsymbol{B}^k = \rho^d \boldsymbol{G}^{-\top} \boldsymbol{N}$, $\rho^d$ is the albedo, $\boldsymbol{N}$ is the surface normal; $\boldsymbol{S}^k = e^k \boldsymbol{G} \boldsymbol{l}^k$, $e^k$ is the light intensity, $\boldsymbol{l}^k$ is the light direction. We denote the object's silhouette normal as $\boldsymbol{N}^s$, the fitted silhouette normal as $\bar{\boldsymbol{N}}^s$, and the predicted silhouette normal by PositionNet as $\bar{\boldsymbol{N}}^s$ that contains GBR ambiguity, *i.e*, $\bar{\boldsymbol{N}}^s = \boldsymbol{G}^{-\top} \boldsymbol{N}^s$.

During training, the output of the LightNet is $\bar{e}^k$ and $\bar{\boldsymbol{l}}^k$. The GBR ambiguity is solved by:

$$\underset{\boldsymbol{G}}{\arg\min} \sum_{k=1}^{f} (|\boldsymbol{I}^k - \bar{\boldsymbol{B}}^{s\top} \bar{\boldsymbol{S}}^k| + \left| \text{Nor} \left( \text{C} \left( \bar{\boldsymbol{N}}^s \right) \right) - \hat{\boldsymbol{N}}^s \right|). \tag{2}$$

Where, $f$ is the number of the light source, $\text{C}(\cdot)$ cuts off the 3rd dimension of $\boldsymbol{n}_k$ (*i.e*, $\text{C}(\boldsymbol{n}_k) \in \mathbb{R}^{1 \times 2}$), and $\text{Nor}(\cdot)$ is the vector normalization operation. This can be easily extended to the non-Lambertian case.

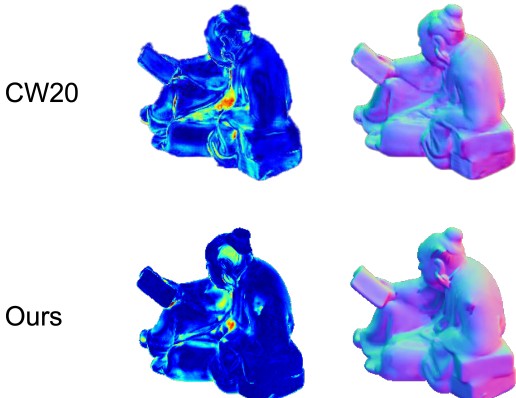

Figure 24: Visual illustration of error maps of the normal. From left to right: error maps of READING, normal map of READING. From top to bottom: ours result and CW20 [4]'s result

Table 12: Quantitative comparison in terms of mean angular error for light direction and scale-invariant error for intensity on APPLE & GOURD [1] and LIGHT STAGE DATA GALLERY [2].

| Model | APPLE | | GOURD1 | | GOURD2 | | AVG | | STANDING KNIGHT | | HELMET | | AVG | |
|---|---|---|---|---|---|---|---|---|---|---|---|---|---|---|
| | dir. | int. | dir. | int. | dir. | int. | dir. | int. | dir. | int. | dir. | int. | dir. | int. |
| YS97 [10] | 25.71 | 0.400 | 22.23 | 0.329 | 29.30 | 0.347 | 25.75 | 0.359 | 37.48 | 0.533 | 34.43 | 0.476 | 35.96 | 0.505 |
| PF14 [7] | 6.68 | 0.109 | 21.23 | 0.096 | 25.87 | 0.329 | 17.92 | 0.178 | 33.81 | 1.311 | 25.40 | 0.576 | 29.61 | 0.944 |
| CH19 [3] | 9.31 | 0.106 | 4.07 | 0.048 | 7.11 | **0.186** | 6.83 | 0.113 | 11.60 | 0.286 | 6.57 | 0.212 | 9.09 | 0.249 |
| CW20 [4] | 10.91 | 0.094 | 4.29 | 0.042 | 7.13 | 0.199 | 7.44 | 0.112 | **5.31** | 0.198 | **5.33** | 0.096 | **5.32** | 0.147 |
| **Ours** | **2.65** | **0.011** | **1.76** | **0.029** | **3.21** | 0.230 | **2.54** | **0.090** | 13.38 | **0.189** | 8.12 | **0.082** | 10.75 | **0.135** |

## 17 Scale on the Input Images

The intensity of 96 images from READING are scaled by 96 variables generated by the the uniform distribution $\mathcal{U}(0.01, 1)$, respectively. The new images are used as the training data to train our method. The results on READING is shown in Fig. 24, which further illustrates that our method can handle the varying light intensity and is free from data bias. While CW20 [4] (15.84 v.s. ours 11.05) fails because they have a pre-defined range on the intensities. We will provide results for all the objects in DiLiGenT in the next round of discussion if the reviewer is still interested.

## 18 Revision of Table 3 in the Main Paper

Table 12 is a revised version of Table 3 in the paper. We highlight the methods that perform the best, and the numbers which are incorrect in our main paper due to a copy-and-paste error are indicated as red in the table. Moreover, the quantitative results in terms of mean angular error for surface normal, light direction, and scale-invariant error for intensity on DiLiGenT benchmark [8] of 5 random tests are shown in Table 13.

Table 13: Quantitative results in terms of mean angular error for surface normal, light direction, and scale-invariant error for intensity on DiLiGenT benchmark [8]. This table summarizes 2 version. Row 2-4 are the reported results in the main paper, row 5-7 are the mean of 5 random tests.

| | | BALL | BEAR | BUDDHA | CAT | COW | GOBLET | HARVEST | POT1 | POT2 | READING | AVG | STD |
|---|---|---|---|---|---|---|---|---|---|---|---|---|---|
| | norm. | 1.15 | 4.41 | 8.78 | 5.08 | 6.14 | 9.49 | 17.68 | 7.94 | 6.12 | 11.82 | 7.86 | - |
| Ours † | dirs. | 1.69 | 3.96 | 1.73 | 2.92 | 4.98 | 6.82 | 7.06 | 3.33 | 3.71 | 7.45 | 4.37 | - |
| | ints. | 0.03 | 0.01 | 0.03 | 0.02 | 0.05 | 0.04 | 0.03 | 0.13 | 0.03 | 0.04 | 0.04 | - |
| | norm. | 1.17 | 4.49 | 8.73 | 4.89 | 6.27 | 9.53 | 18.31 | 7.08 | 5.85 | 12.02 | 7.83 | 0.44 |
| Ours ‡ | dirs. | 1.79 | 3.54 | 2.33 | 2.60 | 5.81 | 8.45 | 7.40 | 3.73 | 2.10 | 7.91 | 4.57 | 0.77 |
| | ints. | 0.01 | 0.01 | 0.03 | 0.02 | 0.20 | 0.04 | 0.03 | 0.07 | 0.04 | 0.05 | 0.05 | 0.03 |

†Reported version in the main paper.
‡Mean of 5 random tests.

## 19 Multiple Specular Basis

We extend the number of the basis used in our SpecularNet from 1 to 9 and switch our rendering model to be identical to [6]. Specifically,

$$m_{ij} = e_j(\rho_i^d + \mathbf{c}_i^\top D(\boldsymbol{h}_{ij}, \boldsymbol{n}_i)) \max\left(\boldsymbol{n}_i^\top \boldsymbol{l}_j, 0\right). \tag{3}$$

Where, $m_{ij}$ is the pixel value, $e_j$ is the light intensity of light $j$, $\rho_i$ is the diffuse reflectance at point $i$, $\boldsymbol{c}_i \triangleq (c_0, ..., c_k)$ is the specular weights generated by the PositionNet, $k$ is the number of basis, $\boldsymbol{h}_{ij}$ is the half-vector between the surface normal at point $i$ and the light direction $\boldsymbol{l}_j$, $\boldsymbol{n}_i$ is the surface normal of point $i$). The results are shown in Table 14. Although there are improvements in objects like BEAR, POT1, and READING, the overall performance drops significantly. This further illustrates that our rendering model is a necessary compromise for the complexity of UPS problem.

Table 14: Quantitative results in terms of mean angular error for surface normal, light direction, and scale-invariant error for intensity on DILIGENT benchmark [8]. 'Ours(9 basis)' indicates using 9 specular basis in SpecularNet.

|  |  | BALL | BEAR | BUDDHA | CAT | COW | GOBLET | HARVEST | POT1 | POT2 | READING | AVG |
|---|---|---|---|---|---|---|---|---|---|---|---|---|
| | norm. | 1.15 | 4.41 | 8.78 | 5.08 | 6.14 | 9.49 | 17.68 | 7.94 | 6.12 | 11.82 | 7.86 |
| Ours | dirs. | 1.69 | 3.96 | 1.73 | 2.92 | 4.98 | 6.82 | 7.06 | 3.33 | 3.71 | 7.45 | 4.37 |
| | ints. | 0.030 | 0.010 | 0.032 | 0.021 | 0.050 | 0.040 | 0.032 | 0.134 | 0.028 | 0.042 | 0.042 |
| | norm. | 1.69 | 4.03 | 12.84 | 5.90 | 8.89 | 14.45 | 20.02 | 6.82 | 9.64 | 11.34 | 9.56 |
| Ours (9 basis) | dirs. | 2.50 | 3.01 | 7.34 | 4.94 | 8.69 | 21.53 | 12.60 | 3.88 | 6.97 | 6.41 | 7.79 |
| | ints. | 0.011 | 0.019 | 0.025 | 0.036 | 0.074 | 0.072 | 0.059 | 0.026 | 0.030 | 0.051 | 0.040 |