# OpenReview forum: "NeIF: Representing General Reflectance as Neural Intrinsics Fields for Uncalibrated Photometric Stereo"
_NeurIPS.cc/2022/Conference — NeurIPS 2022 Submitted_

### Official Review · Reviewer_UR1t · 2022-07-09

**Rating:** 6
**Confidence:** 3
**Soundness:** 3 good
**Presentation:** 3 good
**Contribution:** 3 good

**Summary:**

An unsupervised method is proposed for uncalibrated photometric stereo. It works by parameterising lighting, reflectance, normal map and shadows by neural nets, a.k.a. neural intrinsic fields a.k.a. NeIF. These networks are supervised by rendering loss, with additional regularisations on boundary normal directions, lighting directions, BRDF monotonicity and shadows. Experiments show overall superior results than unsupervised or uncalibrated methods.



**Questions:**

What is ours(16) and ours(19) in table 1,2? Are these results with sparse and all input views because I cannot find any explanation in texts?





**Limitations:**

I think there is room for improvement. See my comments above.

**Strengths And Weaknesses:**

Strengths:
- The method is uncalibrated and unsupervised, meaning it does not require any additional information other than input images themselves.
- Although implicit neural nets are not new to PS, the intrinsic decomposition paradigm makes sense and the method is technically sound.
- Experiment results are convincing.

Weakness:
- I think the paper could benefit from being more upfront about its assumptions and limitations. For example, lighting is assumed to be distant, and BRDF being a bivariate one (with half angle and difference angle parameterisation) with a single specular lobe.
- While the BRDF parameterisation does allow variations between specular and diffuse, one could argue the single specular lobe assumption is too restrictive for diverse, spatially varying materials, e.g. an object made with two specular materials with different BRDFs.
- I need more convincing arguments about why and how the bas-relief ambiguity is resolved here. The boundary normal azimuth penalty in eq(6) does not fully disambiguate light/normal directions. However the experiments seem to suggest both normal and light directions are in fact successfully recovered.
- One strength of this paper is that it deals with varying light intensities. However there is no mention to how much light intensity varies in experiments (e.g. diligent dataset). I wonder how the results would change if you apply a random scale to each input images.
- Minor typos in reference style:
    - line 25: [4], [34] -> [4,34]
    - line 76 [bi2022neural]
    - line 185 [8758025]
    - missing citation numbers in figure 3

---

> ### Author Response · Authors · 2022-08-02
> **Response to reviewer UR1t**
>
> ### “More arguments about why and how the bas-relief ambiguity is resolved by Silhouette Constraints.”
> The expression of the ambiguity during training is the synchronous rotation of estimated surface normal and light direction. While we fix the silhouette normal's azimuth angle, these constraints help mitigate the ambiguity by aligning the predicted surface normal to the fitted silhouette normal's azimuth, which suppresses the rotation of surface normal. Once the surface normal stops rotating, the light direction stops too. Please refer to Sec. 16 in $\color{#FF00FF}{RSupp}$ for further explanation.
>
> ### “How the results would change if apply a random scale to each input image”.
> + We apply a random scaling to the images, making some of the light intensities fall out of the defined range of CW20 [Chen 2020]. We find that our method is robust to the random scaling, while CW20 [Chen 2020] fails significantly because of their pre-assumption on the intensities range. Please refer to Sec. 17 in $\color{#FF00FF}{RSupp}$ for more details.
> + Another evidence to support the effectiveness of our method on the unexpected light intensity could be found in Sec. 12.2 in $\color{#FF00FF}{RSupp}$.
>
> ### "What does ours(10) and ours(16) mean in tables 1 and 2?"
> The number in the bracket means how many lights are used in certain methods. We have added the explanation in the caption of each table in the revised version of $\color{#0000FF}{MPaper}$.
>
> [Chen 2020] Guanying Chen, Michael Waechter, Boxin Shi, Kwan-Yee K Wong, and Yasuyuki Matsushita.“What is learned in deep uncalibrated photometric stereo?”In: Proc.European Conference on Computer Vision (ECCV).2020.

---

### Official Review · Reviewer_zHtR · 2022-07-10

**Rating:** 5
**Confidence:** 3
**Soundness:** 3 good
**Presentation:** 1 poor
**Contribution:** 2 fair

**Summary:**

This paper introduces a novel training framework and model design for unsupervised photometric stereo, which is the problem of estimating surface normals of an object given a set of images with uncalibrated lighting positions. The model leverages Neural Intrinsic Fields, which are a set of MLPs which represent shadows, positions, specularity and light, as well as a CNN encoder aimed at aiding the representational power of the system. The training procedure uses a set of domain-specific loss functions and a warm-up strategy aimed at improving stability and generalization. Ablation studies are provided, which show the effectiveness of the model and the training procedure, and comparisons with previous work show quantitative improvements.

**Questions:**

- On line 34, it is claimed that supervised UPS methods fail at generalizing to data points outside the training distribution. The authors thus present unsupervised learning as a better alternative for enhancing generalization. This argument is repeated throughout the paper, but, why is this the case? Why is unsupervised learning inherently better than learning from labels in this particular scenario? Please explain better or demonstrate quantitatively.
- The paper could benefit from referring the reader to important papers that would help in contextualization and learning about this particular domain. For Neural Fields, I would suggest referring to (Neural Fields in Visual Computing and Beyond, Xie et al.), which should help the reader understand that this UPS paper is not that closely related to NeRF papers as the title and the related work suggest. For material models and intrinsic decomposition, I would suggest (A Survey on Intrinsic Images: Delving Deep Into Lambert and Beyond, Garces et al.).
- Please discuss the limitations of the material model in terms of which effects are not modelled with the current image formation assumptions, as many effects cannot be represented by NeIF but this is not clear to the reader.
- Please explain terminology that is not explained nor it is obvious to the reader. For example, GBR, Lambertian objects, etc.
- Please include standard deviations in the tables, to better understand whether or not the improvements are significant.
- Please double-check the references (eg line 185).


**Limitations:**

The paper does not include any negative impact analysis of their work. It is difficult to imagine any direct negative impact of surface normal estimation, but I would argue that this method is computationally very expensive at test time and it may be detrimental in terms of energy consumption. In terms of limitations, the paper includes an adequate limitation analysis. To complete it, I would suggest the authors to discuss the limitations of the material model that their method assumes for training, as it does not account for complex light phenomena, including sub-surface scattering, transmittance or iridescence.

**Strengths And Weaknesses:**

*Summary*

This paper provides valuable ideas for training methods for unsupervised photometric stereo, which is an important problems in computer vision and graphics. The proposed method is sound, ablation studies are provided that validate the effectiveness of the design choices of their method, and the model combines advances in machine learning with domain-specific ideas in a smart way. However, the paper is not well written, presentation is arguably poor and comparisons with previous work do not show a huge improvement. Given that the paper is hard to read and comparisons with previous method do not show significant improvements, it is difficult to recommend this paper for acceptance at its current state.

Strenghts:

- This paper introduces novel ideas for uncalibrated photometric stereo, combining advances in machine learning with ideas from more classical computer vision and graphics.
- The proposed method is technically sound, and ablation studies show the impact of each component in the overall generalization capabilities of the model. Evaluation is somewhat exhaustive.
- Comparisons with previous work are complete and exhaustive to the best of my knowledge.

Weaknesses:

- The paper needs revision in the way it is presented and written. It is hard to follow and some sentences are very difficult to understand. The paper would benefit strongly from a revision on this level. Further, it is presented in a way that assumes a lot of previous knowledge from the reader. Please see below for suggestions on references that may help the render understand better this paper and its context.
- Previous work is not adequately studied and it is not clear where this paper sits on the literature. By reading the title and some parts of the paper, it would seem that this is a NeRF paper with applications for uncalibrated photometric stereo. This is not the case, and I believe the paper would benefit from a more clear distinction with Radiance Fields methods.
- Comparisons with previous work could be presented better, in my opinion. For example, standard deviations should be shown next to averages (eg Table 1) and sometimes previous work shows better perfomance that the proposed method, but the authors decide not to highlight that (eg Table 3, CW20 is better in AVG dir but this is not in bold). It is not entirely clear how much better the proposed method is with respect to previous work, and, when it is not, why previous work is better.

---

> ### Author Response · Authors · 2022-08-02
> **Response to reviewer zHtR**
>
> ### “The distinction between our methods and the Radiance Fields methods.”
> Our method follows a similar idea to NeRF, which implicitly represents intrinsic components for radiance reconstruction. Their differences are as follows.
> + **View and light.** Multi-view Stereo varies in the view directions with the static light, while Photometric Stereo varies in the light directions with a fixed view direction.
> + **Rendering processes.** Because of that difference, we have different rendering processes. That is, NeRF uses volume rendering (dependent on the view direction) while we use physical-based rendering (dependent on the light direction).
> + **Application.** NeRF is originally applied to view synthesis, while ours is expected to be applied to relighting. Considering the narrow application scenarios of the relighting under directional light, we focus on the accurate estimation of surface normal, which is the main target of photometric stereo.
> + **Input.** NeRF's sampling process heavily relies on the input of camera's poses, while for the calibrated PS problem, the intensity and direction of the light are also imperative as the input. Just like recent works that try to predict the camera's poses and relax the input of NeRF (e.g., [Su 2021]), we extract extra clues from shadow and specular to estimate the light's condition and address the problem of UPS.
>
> We will add the above discussions to the final version of our paper (in Sec. 2).
>
> ### “We do not highlight the numbers when other methods perform better than us.”
> We apologize for the misleading and confusion. It was a copy-and-paste error for the average. The updated numbers (highlighted in red) for Table 2 are shown in Table 12 in $\color{#FF00FF}{RSupp}$. We also include the standard deviation of objects in DiLiGenT [Shi 2016]  dataset for reference.
>
> ### “Why is unsupervised learning inherently better than supervised methods.”
> + In [Ren 2022], the authors point out that current supervised methods overfit the DiLiGenT dataset [Shi 2016], and unsupervised methods like TM18 [Taniai 2018] and the Least Square are much more stable than the supervised methods.
> + Supervised methods like CW20 [Chen 2020] have a pre-defined range for the light intensity, and there are domain gaps between the training set and the testing set. To further verify this, we apply a random scaling to the images, making some of the light intensities fall out of the pre-defined range of CW20 [Chen 2020]. We find that our method is robust under the random scaling, while CW20 [Chen 2020] fails because of their assumption of the intensities range (results can be found in Sec. 17 in $\color{#FF00FF}{RSupp}$). We also conduct experiments on our newly collected data, where the light is not assumed to be ideally directional. Results in the experiments also indicate that supervised methods suffer from training data bias and are sensitive to the data assumption (results can be found in Sec. 12 in $\color{#FF00FF}{RSupp}$).
>
> ### “The standard deviation of the experiments should be provided.”
> We provide the standard deviation and the mean value for 10 objects on the DiLiGenT benchmark [Shi 2016] under 5 random tests in Sec. 18, $\color{#FF00FF}{RSupp}$.
>
> ### “Explain the terminologies.”
> We will add a more comprehensive explanation of the terminologies in the final version.
>
>
> [Shi 2016] Boxin Shi, Zhe Wu, Zhipeng Mo, Dinglong Duan, Sai-Kit Yeung, and Ping Tan.“A benchmark dataset and evaluation for non-lambertian and uncalibrated photometric stereo”.In: IEEE Transactions on Pattern Analysis and Machine Intelligence (TPAMI) .2016.
>
> [Taniai 2018] Tatsunori Taniai and Takanori Maehara.“Neural inverse rendering for general reflectance photometric stereo”. In: Proc. International Conference on Machine Learning (ICML). 2018.
>
> [Chen 2020] Guanying Chen, Michael Waechter, Boxin Shi, Kwan-Yee K Wong, and Yasuyuki Matsushita.“What is learned in deep uncalibrated photometric stereo?”In: Proc.European Conference on Computer Vision (ECCV).2020.
>
> [Su 2021]Shih-Yang Su and Frank Yu and Michael Zollhofer and Rhodin Helge, “A-NeRF: Articulated Neural Radiance Fields for Learning Human Shape, Appearance, and Pose”. In: Proc. Conference on Neural Information Processing Systems (NeurIPS). 2021.
>
> [Ren 2022] Jieji Ren, Feishi Wang, Jiahao Zhang, Qian Zheng, Mingjun Ren, and Boxin Shi, “DiLiGenT102: A Photometric Stereo Benchmark Dataset with Controlled Shape and Material Variation”. In: Proc. Computer Vision and Pattern Recognition (CVPR). 2022.)

---

> ### Comment · Reviewer_zHtR · 2022-08-07
> **Post-rebuttal comments**
>
> I appreciate the thorough respose of the authors to my concerns and to those of other reviewers. I am confident the paper can improve after reading the authors' rebuttal and I am therefore willing to slightly increase my rating.
>
> However, despite the efforts in improving the motivation and fixing typos, I think the paper lacks clarity in its writing and its discussion of previous work.

---

> > ### Author Response · Authors · 2022-08-07
> > **Reply to Reviewer zHtR**
> >
> > Dear Reviewer zHtR,
> >
> > We sincerely thank you for your support and further suggestions on the clarity. The revised version we submitted (finished a few weeks ago) did not include the discussion of rebuttal. We list a summary of the revision for the final version of our paper for your consideration.
> > + Sec. 2.1: The discussion about the difference between our works and previous neural field works will be added.
> > + Sec. 2.2: A detailed discussion and comparison with different materials models used in previous works will be added.
> > + Sec. 2: The work of intrinsic decomposition will be discussed in a new subsection.
> > + Sec. 3.1: More explanation about the terminologies, for example, GBR ambiguity and Lambertian objects, will be added.
> > + All references mentioned in the rebuttal will be included in adquetate places to enhance the overall clarity.
> >
> > If you have any further suggestions about improving the clarity of our paper or the discussion with previous work, please do not hesitate to let us know, and we are happy to include them.

---

> ### Comment · Reviewer_zHtR · 2022-08-08
> **Post-Rebuttal Rating**
>
> After reading the authors response, and the other reviews, I increase my rating to a borderline accept.

---

### Official Review · Reviewer_yGxh · 2022-07-10

**Rating:** 3
**Confidence:** 5
**Soundness:** 2 fair
**Presentation:** 3 good
**Contribution:** 1 poor

**Summary:**

This paper proposed a method that aims to solve the uncallibrated photometric stereo problem where the input light intensities and light directions are unknown. The key idea of this paper is to represent the surface normal, and surface reflectance, shadows, and lights as neural networks. And then jointly train those neural networks via the image reconstruction loss in an unsupervised manner.

The proposed method explicitly considers the clues from specular effects and cast shadows for solving this problem.
Specifically, the paper proposed: a PositionNet for outputting the surface normal, specular weights, and diffuse weights of the object; a SpecularNet that takes half angle and surface normal as input, outputting the specular component; a ShadowNet that explicitly predict the shadow map; and LightNet that takes RGB images as input, outputting the light intensities and directions.

The performance of the method was evaluated on DiLiGenT and some other real datasets and achieved lower normal estimation errors than other state-of-the-art algorithms.

**Questions:**

How will the number of input images affect the testing time of this method? e.g., how long does it take to test with 16 images at input?

**Limitations:**

- The rendering equation doesn't take inter-reflection into account. Hence, in theory, the proposed method will fail if the inter-reflection effects dominate the surface.
- The photometric stereo images are usually taken in a dark room with limited lighting exposures, which may also introduce high-level image noise. I encourage the authors to discuss more about the robustness of this method when the image assumptions are violated in these aspects.

Suggestion:
- I encourage authors to list the testing time of other most recent inverse rendering-based methods [44,25,26] for comparison.

**Strengths And Weaknesses:**

Strengths
- The paper is well written and easy to follow.
- The evaluation of the proposed method is extensive and complete, and supports the claimed effectiveness of each component of the method.



Weaknesses
- The design of the PositionNet and SpecularNet is very similar to the design of [26] ("Neural Reflectance for Shape Recovery with Shadow Handling"). But I didn't find a proper reference to [26] to state the difference: both [26] and this work have a network outputting the surface normal, diffuse albedo, and specularity; both [26] and this work have a SpecularNet which take normal, half-angle, and view direction and outputting the specular effects of the surface; both [26] and this work explicitly compute the shadow map which is then used in image rendering; both [26] and this work use a warm-up loss for early-stage supervision of the shadow map. I encourage the authors to discuss the differences to help clarify the position of this work.

- How does the SpecularNet work for objects with specular-SV-BRDF? Especially for a object with two different materials: soft and shiny specular effects? (such as a object make up with two different metals: one with high roughness, one with low roughness).
The input of SpecularNet is only the surface normal, half angle, and view direction. So for different positions over the surface, they will have the same SpecularNet, which means that it assumes all the surface points will have the same specular effects (same specular BRDF curve, or same specular roughness). But this isn't the case for most of the objects in the world: for some objects with spatially-varying BRDF where both soft specular effects and very shinny and sharp specular effects exist on the surface.

- Unsupervised or weak-supervised? The method is claimed to be an unsupervised method, but as mentioned in line 194, the method uses the light estimation results from YS97 as a weak-supervision signal to train the LightNet. Besides, from Supplementary Table 9, it seems that without using YS97 as weak-supervision in the early-stage warm-up will cause a significant drop in accuracy.

- For the design of LightNet, since we need to train the LightNet in a weak-supervised manner from YS97, why not just use light directions from YS97 as the initialization and directly optimize on these lights. (i.e. why not explicitly optimized the light directions instead of optimizing the parameters of LightNet?)

- Since we need to train the LightNet in a weak-supervised manner, why don't we choose a better method as the "teacher network"? For example, why not use light direction and intensities from SDPS[12] as the guidance to guide the training of LightNet?

- The ambiguity in ShadowNet. I believe that there is ambiguity in the design of the proposed ShadowNet. From equations (3) and (4), it seems that the authors aim to use ShadowNet to represent the cast shadows. In the object's appearance, both pixels that are in cast-shadows and attach-shadows have 0 intensity.
But their causes are completely different; the cast shadows are caused by objects self-occluding the light source; the attach-shadows are caused by the surface back-facing the light source. And surfaces with extremely dark materials may sometimes also have zero pixel intensity under certain light directions. And the proposed method will have ambiguity under the above situations. The downside of this ambiguity can actually be seen in Fig 4, 5th column, where the generated shadow map shows the inconsistent boundaries in many objects (e.g Standing K. and Gourd2, Apple).

---

> ### Author Response · Authors · 2022-08-02
> **Response to reviewer yGxh**
>
> ### “Unsupervised or Weak-supervised.”
> We define our method as “unsupervised” due to the following reasons:
> + YS97 [Yuille 1997] is an unsupervised method. Besides, we only use the estimated azimuth of the light to initialize the weights of the LightNet at the warm-up stage (10 epochs). We drop it for the rest of the training procedure (490 epochs), so the effect of the YS97 [Yuille 1997] is very limited. Kindly note that the performance of YS97 [Yuille 1997] for light estimation is less competitive (see Table 2 in Sec. 4.2 in $\color{#0000FF}{MPaper}$).
> + [Taniai 2018] used Least Square as weak supervision at their warm-up stage, but they classified their method as an unsupervised learning method. Therefore, we define ours as the unsupervised method as well.
>
> ### “Why we do not use the result predicted by YS97 [Yuille 1997] and optimize it.”
> There are two reasons.
> + We use results from LightNet instead of those from YS97 [Yuille 1997] because we hope to continuously exploit the mutual information conveyed in the observed images, which serve as the input of LightNet. In contrast, dropping LightNet and using the results from YS97 [Yuille 1997] cannot persistently exploit this mutual information during training. The mutual information establishes a neural light field that implicitly records the scene's light condition. According to the correlation study in Sec. 8 in  $\color{#FF7D00}{MSupp}$, we experimentally find that LightNet infers the light condition from the light code extracted by the CNN-encoder, which contains abundant specular and shading information and highly correlates to the ground truth light condition. The shading and specular information are discarded in the suggested solution based on YS97 [Yuille 1997].
> + YS97 [Yuille 1997]’s result is not accurate, especially in sparse light conditions like Ball. It doesn't converge to an appropriate estimation (90 degrees of MAE for the Ball). Therefore, it would be risky to implement it directly into the network.
>
> ### “Why we do not use the result of a better method to supervise the training of the LightNet.”
>
> We do not use the result of a better method because we want to highlight our unsupervised learning manner. As can be found from Table 2 in $\color{#0000FF}{MPaper}$, better methods are achieved in a supervised learning manner and could suffer from training data bias. We show how such methods fail when the light is non-ideal directional light while our method could still achieve reasonable estimation. Please refer to Sec. 12 in $\color{#FF00FF}{RSupp}$ for preliminary experimental evidence.
>
> ### “There are ambiguities between the cast shadow and the attached shadow, as well as between the shadow and the point with low pixel intensity, which contributes to an inconsistent boundary in many objects.”
>
> We agree that there are artifacts in the predicted shadow map due to the listed ambiguities, which are hard to overcome. The per-pixel prediction of ShadowNet is the main reason for generating inconsistent boundaries. We use a per-pixel prediction manner rather than that in [Li 2022] because
> + we need the backpropagation from the ShadowNet to the LightNet for robust light estimation;
> + the per-pixel manner leads to the differentiable ShadowNet for backpropagation.
>
> ### “Results on noisy images taken with limited lighting exposures.”
> The image's noise will degrade our model's performance and other inverse rendering methods. Fortunately, there are many efficient ways to avoid underexposures, e.g., increasing the light source's intensities or enabling exposure compensation in the photographic equipment. Due to the limited time allowed during the rebuttal period, we are not able to experiment on such data. We will provide results in the next round of discussion if the reviewer is still interested.
>
> [Yuille 1997] Alan Yuille and Daniel Snow. “Shape and albedo from multiple images using integrability”. In: Proc. Computer Vision and Pattern Recognition (CVPR). 1997.
>
> [Li 2022] Junxuan Li and Hongdong Li. “Neural Reflectance for Shape Recovery with Shadow Handling” In: Proc. Computer Vision and Pattern Recognition (CVPR). 2022.
>
> [Taniai 2018] Tatsunori Taniai and Takanori Maehara.“Neural inverse rendering for general reflectance photometric stereo”. In: Proc. International Conference on Machine Learning (ICML). 2018.

---

> ### Comment · Reviewer_yGxh · 2022-08-07
> **Post-rebuttal comments**
>
> After carefully reading the authors' responses and other reviewers' comments, I still believe that this paper is far from ready to be published and presented in NeurIPS.
>
> In related work section, I appreciate the thoughtful comparison between this work and [Li 2022]. It really helps to clarify the position of this work in this area.
> The authors are encouraged to include this comparison in the related work (Sec 2) and method (Sec 3) sections.
> Additionally, as mentioned by Reviewer zHtR, the discussion about the difference between neural-field-works, GBR ambiguity, and material modeling also needs to be included in the paper.
> I think that there is still a significant revision and modification need to be made in the current revised version of the paper.
>
> In the method part, the assumption and limitations of the method need to be added to the paper.
> That the method use only 1 specular basis with only gray-level intrinsics, which will lose the ability to model materials with different specular lobes (AKA specular roughness) and different colors.
>
> In the LightNet part:
> I think it is not very scientific to claim that: [in order to claim this method as an unsupervised method, we chose inaccurate lightings as the weak-supervision signal]. My point of view here is that it is totally acceptable to use a more accurate state-of-the-art light estimation method as the weak supervision signals for the LightNet, and claim this paper as the weak-supervised one. As both of the above choices need lighting estimation input from other methods.
> And I still believe that, although the deep-learning-based light estimation methods may suffer from domain gaps, they are still the state-of-the-art methods in most scenarios. Besides, the YS97 fails in many objects (as shown in Table 2, object Ball, Harvest). I would suggest the author to conduct more experiments to validate their choices on choosing the weak-supervision light estimation siginals.
>
>
> In the experiment section: since this paper focuses on a per-object optimization problem, the total time cost during the entire optimization stage should be compared rather than only the "testing stage". As mentioned by author that the method takes ~4 hours on RTX 2080, compared to previous methods: ~1 hour [Taniai 2018], ~1 hour [Kaya 2021] and ~6 mins [Li 2022], the proposed method is significantly slower.
>
> In the experiment section: "Results on noisy images" can be simply achieved by adding a different level of gaussian noise to the input RGB images and testing how the performance is affected by the noise level. And compare it with previous deep-learning-based methods to see the robustness of image noises.
> Since the author also admitted that noisy input would lead to degrading the performance, I also suggest the authors to include it as the limitations of the paper.
>
> In summary, while I do appreciate the thoughtful explanations and extensive evaluations the authors provided during the rebuttal period, I believe that this paper is far from ready to be published. The authors are encouraged to modify and adjust the paper in most of the sections as all reviewers suggested. And currently, the revised version of this paper has already reached the 9-pages limitation (see "https://openreview.net/pdf?id=Qr8n979lusV"). It is quite challenging and has a lot of uncertainty about whether their revised main paper can adequately address all the essential comments and suggestions from all reviewers.
>
> Hence, I keep my rating as rejection.

---

> > ### Author Response · Authors · 2022-08-07
> > **Response to the post-rebuttal comments from Reviewer yGxh**
> >
> > Dear Reviewer yGxh,
> >
> >  We sincerely thank your suggestions and respect your rating. According to the post-rebuttal comments, your primary concern is the **uncertainty about the revision**. We will definitely and strictly follow your and other reviewers' suggestions to revise the manuscript. The following facts can support our capacity and strong determination for your consideration.
> >
> > + **Certainty of results.** We have provided **all code, weights, and data** used in our experiments for reproduction and do not hide any trick or unclarity during our paper submission and rebuttal stages. This fact could guarantee the revision to be certain regarding the results and experiments.
> > + **Certainty of revision that follows your suggestions.** We have tried our best to **point-wisely** conduct *"thoughtful explanations and extensive evaluations"*  according to suggestions from you and other reviewers, even in the rushing rebuttal period. Considering there is a much longer time for us to revise the final manuscript, the certainty of revision that is **point-wisely** based on your suggestions could also be guaranteed.
> > + **Certainty of writing.** Reviewer yGxh (you) and Reviewer U25q both acknowledge that our preliminary submission is easy to follow or understand. This fact could guarantee the revision to be certain regarding the quality of our writing.
> > +  **Certainty of paper length**. We are fully aware of the limitations on pages in NeurIPS. Therefore, our final version will first guarantee the overall clarity and completeness of the paper. To ensure we have enough space for the new contents, we will slightly shorten some of our previous sections and move those contents to the supplementary material. (e.g., the results and analysis about sparse UPS in Sec 4.2 can be much briefer than the current version; some less representative methods/samples in Table 1/Figure 4 and Table 3/Table 4 will be moved to our supplementary material). We promise to submit the supplementary material as the formal file in the camera-ready version and include all discussions and experiments.
> >
> > If you have any concerns, please do not hesitate to let us know. We are willing to take your suggestions and respect your determination.

---

### Official Review · Reviewer_U25q · 2022-07-11

**Rating:** 5
**Confidence:** 5
**Soundness:** 3 good
**Presentation:** 3 good
**Contribution:** 3 good

**Summary:**

This paper presents the neural uncalibrated photometric stereo method under the single directional lighting setup. Following recent neural-field-based algorithm, the method simultaneously predicts surface attributes and lighting parameters in a unified self-supervised learning method. Shadows are explicitly considered for filtering out its contribution in the optimization process. In the evaluation, the proposed method achieved state-of-the-art performance on the public DiLiGenT benchmark.

**Questions:**

- The concurrent work [1] also introduced the neural fields for the photometric stereo. This work and [1] seem very close to each other for the modeling of shadow and BRDF even though this work solves the uncalibrated task unlike [1]. Other than this difference, is there any critical difference between these works that makes this work more significant?

- The proposed method certainly works in the environment assumed in the benchmark, but there is some question as to whether this research direction is the right for the photometric stereo field. Since many applications in photometric stereo are time-critical (i.e., surface defect inspection), it is questionable whether the small improvement in accuracy obtained with the neural field's approach is worth the sacrifice the efficiency in learning-based photometric stereo methods which only take less than a second to recover comparable results. In addition, I believe that data-driven methods can handle a more generic environment including the global illumination effects under a more complicated lighting environment than an inverse rendering that relies on a specific physically-based model. There is no point in simply improving benchmark scores if they are not immediate to real problems. Therefore, I would like to see a little more convincing motivation of this work.


**Limitations:**

The limitation is addressed in the paper.

**Strengths And Weaknesses:**

Strength:
- The proposed method is very clearly written and the algorithms are easy to understand.
- The benchmark study shows that the proposed method is superior to the competitor.
- It is very interesting that the loss is propagated from the cast shadow since the shadow mapping is generally indifferentiable.
- Ablation studies have been conducted for each of the technical components.
- The application of neural fields to uncalibrated photometric stereo is new among published studies.

Weakness:
- It seems an overclaim that there are no explicit assumptions about the light and reflectance since the light is obviously constrained to the single directional lighting excluding near and natural lighting.  As for the reflectance, how to handle spatially-varying BRDF is not clear since SpecularNet is only queried by nh and vh without the location of the pixel. It seems that the proposed method only scales a single BRDF by the specular coefficient rho_k, which obviously limits its representability. The more proper manner seems to define multiple bases (neural networks) and compute multiple coefficients as was done in [1].

- The behavior of ShadowNet is not clear to me. Since the gradient of binarization layer is static, the loss doesn’t seem to be propagated. Therefore, it seems to me that the shadow is not explicitly considered for recovering the geometry unlike the shape-from-shadow and it is only used to detect outliers to make the reconstruction loss robust to cast shadows as is similar with [1]. If so, it seems that simply detecting shadows statistically and reflecting them in the loss function would be sufficient. This work may need evidence that results from ShadowNet are superior to statistical shadow removal.

- The physical parameters recovered by the proposed method have not been properly evaluated, so it is not possible to examine whether the formulation is appropriate. From the results presented in Fig17 of supplementary, it appears that the proposed method couldn’t successfully recover physical attributes such as shadows and reflectance, which makes the proposed method less useful.

- All results are based on public evaluation data sets taken in carefully controlled environments. In addition to those results, I would have liked the authors to have shown the actual results taken by authors to demonstrate its availability in real use-case.

[1] Li and Li, Neural Reflectance for Shape Recovery with Shadow Handling, CVPR2022

---

> ### Author Response · Authors · 2022-08-02
> **Response to reviewer U25q**
>
> ### "The loss of the ShadowNet does not seem to backpropagate, and it will be more efficient to use statistical methods for shadow handling instead of the ShadowNet."
>
> Our ShadowNet does backpropagate because the gradient of the Binary Layer is 1, and it works like a linear activation function when the loss is backpropagated. This backward path is imperative to our framework's light estimation and the normal map. Without that, the performance degrades significantly. Please refer to Sec. 11 in $\color{#FF7D00}{MSupp}$. We further conduct additional experiments to compare with results from the method using the statistical information as the supervision of shadow. The results show the necessity of the ShadowNet. Please find the detail in Sec. 15 in $\color{#FF00FF}{RSupp}$.
> However, the backpropagation from reconstruction loss to ShadowNet results in the inconsistent boundary (as figured out by Reviewer yGxh) because of the per-pixel estimation manner.
>
> ### "The physical parameters recovered by the proposed method have not been properly evaluated, so it is not possible to examine whether the formulation is appropriate. The proposed method couldn’t successfully recover physical attributes, which makes the proposed method less useful."
>
> Due to the ambiguity between the light intensity and the BRDF parameters, UPS methods can hardly solve all physical parameters without any prior, especially in an unsupervised manner. We have to simplify the model or computation of BRDF, i.e., by using a single lobe for specular reflectance and computing the cast shadow in a per-pixel manner. However, thanks to the unsupervised learning and joint optimization, our method is robust to training data bias and could better balance the accuracy between estimated light and surface normal. The results in Sec. 12 in $\color{#FF00FF}{RSupp}$ show the superior performance advantage over state-of-the-art methods on data collected in casual scenarios.

---

### Author Response · Authors · 2022-08-02
**Response to all reviewers (part 1/2)**

We sincerely thank all reviewers for their valuable comments and suggestions. We will fix the typos, add more references, and explain more about the terminology in the final version.
For clear reference, we represent the main paper as $\color{#0000FF}{MPaper}$, supplementary material for main paper as $\color{#FF7D00}{MSupp}$, and supplementary material for rebuttal as $\color{#FF00FF}{RSupp}$. These files could be downloaded from either the OpenReivew system or the anonymous link https://drive.google.com/drive/u/1/folders/1XZEb2H3-ZTuTyZaxiewa3DDH-urVbuOk.

The typos issue was partly fixed after our submission, and a reference revision can be found in the revised paper. Kindly note that this revision was completed several weeks ago, and the final version will merge all discussions in this rebuttal.

Below, we first address the common questions. We then respond each reviewer's specific questions in "official comments" to each reviewer. All discussion and results will be merged into our final version of the paper or its supplementary material.


### Comparison with [Li 2022]
By the time we submitted the paper to NeurIPS, the code and the camera-ready version of [Li 2022] were not released. Here, we would like to give a comprehensive comparison between our work and [Li 2022].

+ **Shadow handling**. A primary difference between our work and [Li 2022] is the way of shadow handling, which is one of the main contributions of [Li 2022]. First, we have different implementations. We use the Binary Layer to ensure the training of our ShadowNet is differentiable by minimizing the reconstruction loss for each pixel. In contrast, the reconstruction loss in [Li 2022] **cannot** backpropagate to their depth net. Because they calculate the geometry loss between the predicted surface **normal map** and the **depth map** for backpropagation (The predicted shadow map is very sensitive to the normal map acquired at the 500th epoch as they will fix the depth net in the rest of the epochs once it is well trained separately based on that normal map). Second, our shadow handling module plays a different role from that in [Li 2022]. As reviewer U25q mentioned, [Li 2022] uses the shadow handling module to refine and stabilize the reconstruction results. In contrast, our ShadowNet is jointly trained with other modules. Our ShadowNet not only plays an important role in stabilizing the training but also provides extra clues for the LightNet, thanks to its differentiability regarding the reconstruction loss. Third, because ShadowNet is necessary for our training, its effects are irreplicable. This can be validated by our experiments in Table 9 in $\color{#FF7D00}{MSupp}$. This is also validated by the additional experiment in Sec. 15 in $\color{#FF00FF}{RSupp}$, where we remove ShadowNet, and it performs poorly on light estimation.

+ **Reflectance modeling**. There are two differences: 1) The number of the basis for the specular reflectance is 1 (ours) vs. 9 ([Li 2022]). 2) Our method applies the spatially varying scaling factors to diffuse and specular terms while [Li 2022] only applies to the specular term. The first difference degrades the capacity of our reflectance model. However, it is necessary due to the difficulty of the UPS problem. Because a reflectance model with a smaller number of unknowns helps stabilize the training. This can be validated by our experiments in Table 14 in $\color{#FF00FF}{RSupp}$. The second difference makes our model handle spatially varying BRDF.

+ **UPS vs. PS**. As figured out by reviewer U25q, we solve a different problem of UPS compared with [Li 2022], where we need to optimize the networks to jointly predict light, intrinsic parameters, and the normal map. Different from [Li 2022] that estimate the normal map and the intrinsic parameters with known lights (i.e., without the worries of GBR ambiguity), we have to simplify the computation of specular reflectance and the shadow, aiming at exploiting them as extra clues to stabilize the optimization of the normal map and light. Such simplification inevitably introduces less accurate estimation. To handle the ambiguity between these unknowns, we introduce several additional constraints for the model optimization (Sec. 3.3 in $\color{#0000FF}{MPaper}$), such as gradient penalty and silhouette clues.

[Li 2022] Junxuan Li and Hongdong Li. “Neural Reflectance for Shape Recovery with Shadow Handling” In: Proc. Computer Vision and Pattern Recognition (CVPR).2022.

---

> ### Author Response · Authors · 2022-08-02
> **Response to all reviewers (part 2/2)**
>
> ### Experiments in real use-case under complex light phenomena
> We follow reviewer U25q's suggestion and elaborate on this point by conducting additional experiments. Specifically, we collect data in **"casual" scenarios** with ambient light and non-ideal directional light for experiments (see Fig. 21 in $\color{#FF00FF}{RSupp}$).
> The light definitely dissatisfies the requirement of classic photometric stereo. However, it is more common and could be easily accessed by casual users. Thanks to the unsupervised learning manner and joint optimization, our method is free from training data bias and could balance the performance between light and surface normal estimation, resulting in superior performance advantages over other state-of-the-art methods. Please find the detail in Sec. 12 in $\color{#FF00FF}{RSupp}$.
>
> ### Spatially varying BRDF
> We agree that the capacity of our reflectance model is somewhat restricted by the specular model, which implicitly shares the same specular reflectance model for different points. However, we must emphasize that our model could have spatially varying BRDF by **assigning different scaling factors $k_d\in [0,1]$ to different spatial points**. That is, $k_d$ controls the specularity of a certain point. The reflectance could be Lambertian if $k_d$ is 1 and non-Lambertian otherwise. The evidence could be found from the result of Standing Knight. Please refer to Sec. 13 in $\color{#FF00FF}{RSupp}$ for the visualization of the predicted BRDF.
>
> ### Assumption and limitation on the material model and light
> + Assumption on light. Our method works under the classic setting of uncalibrated photometric stereo (UPS). Therefore, it assumes the single directional light distributing in the upper hemisphere.
>
> + Assumption on the material model. The SpecularNet takes the $\boldsymbol{v}^{\top} \boldsymbol{h_j}$ and $\boldsymbol{n}^{\top} \boldsymbol{h}_{\boldsymbol{j}}$ as the input. So, it assumes the highlight on the object to be single lobe and bivariate.
>
> + Limitation on the material model. We do not consider the inter-reflection and only predict the gray-level intrinsics for simplification. This is because UPS with unsupervised learning is much more challenging, and we have to simplify the specular and shadow model for stable training. Consequently, we assume the specular reflectance to be isotropic, so it cannot handle the object with anisotropic materials (e.g., the glass bead in Venus, shown in Fig. 21 in $\color{#FF00FF}{RSupp}$).
>
>
> ### Elaborate our motivation
> We follow the suggestion by reviewer U25q and experiment on our newly collected data. We collect three objects with environments that are not carefully controlled. The results directly illustrate that inaccurate light dramatically degrades the performance of comparison methods (CW20 [Chen 2020], LL22 [Li 2022]) due to the accumulating errors, as we state in line 36 in $\color{#0000FF}{MPaper}$. In contrast, our method is more robust due to unsupervised learning and light-normal joint optimization. Please refer to $\color{#FF00FF}{RSupp}$ for more details.
> We will **rewrite the motivation** to highlight the role of our unsupervised learning and light-normal joint optimization, merge the results and discussion, and release the code and all data to the public for reproduction.
>
> ### Concerns about the test time.
> + The average test time for objects in DiLiGenT is 0.69 s, 1.13 s, 0.79 s for our method, [Li 2022], and [Taniai 2018], respectively, given 96 observed images. Thanks to parallel computing, per-pixel methods (i.e., ours, [Li 2022]) could deal with different points simultaneously and have similar or even better computation efficiency as compared with all-pixel methods (i.e., [Taniai 2018]). Details can be found in Table 10 in $\color{#FF00FF}{RSupp}$.
>
> + The average training time for our method is about 4 hours on RTX 2080. The slow training time of our method is that our depth reconstruction algorithm runs **on the CPU**. We will release the GPU version of our implementation and provide the running time comparison in the next round of discussion if the reviewer is still interested.
>
> [Chen 2020] Guanying Chen, Michael Waechter, Boxin Shi, Kwan-Yee K Wong, and Yasuyuki Matsushita.“What is learned in deep uncalibrated photometric stereo?”In: Proc.European Conference on Computer Vision (ECCV).2020.
>
> [Li 2022] Junxuan Li and Hongdong Li. “Neural Reflectance for Shape Recovery with Shadow Handling” In: Proc.Computer Vision and Pattern Recognition (CVPR).2022.
>
> [Taniai 2018] Tatsunori Taniai and Takanori Maehara.“Neural inverse rendering for general reflectance photometric stereo”. In: Proc. International Conference on Machine Learning (ICML).2018.

---

### Author Response · Authors · 2022-08-07
**Looking forward to your post-rebuttal comments and suggestions**

Dear AC and all reviewers:

We sincerely thank you for all the comments and suggestions that help us enhance the completeness and persuasiveness of our paper.

Since there are two days left for the rebuttal phase, we have not yet heard any post-rebuttal response.

Any advice from you is valuable to us, and we cherish your comment as a chance to improve our work further. If there are any additional experiments or clarifications we can offer, please don't hesitate to let us know, as we would like to convince you of the merits of our paper.
Thank you again for your time on our paper, and looking forward to your response!

To view the comments, please check the reviewers console.

To view the related supplementary material, please check: https://drive.google.com/drive/u/1/folders/1XZEb2H3-ZTuTyZaxiewa3DDH-urVbuOk

To view the pre-trained models and codes we used on the new dataset, please check: https://drive.google.com/drive/u/1/folders/1XZEb2H3-ZTuTyZaxiewa3DDH-urVbuOk

---

### Meta-Review · Area_Chair_8HyX · 2022-09-01

**Recommendation:** Reject
**Confidence:** Less certain

**Metareview:**

This paper had reviews ranging from a Reject to a Weak accept.

The key shared concerns among reviewers were concern about how much is really new relative to the CVPR paper [Li et. al, 2022].  The most negative reviewer (who is quite expert in this field) engaged strongly in the discussion with the authors, highlighting sustained concerns about novelty, substantially slower speed at rendering time, and the loss of the ability to render materials with interesting reflectance properties.

I found this review most accurate and detailed.  The remaining reviews, while borderline or weakly positive, retained concerns about the limited lighting and reflectance properties.

Therefore I am deciding to reject this paper

**Award:**

No

---

### Decision · Program_Chairs · 2022-09-14

Reject